# Fatty acids homeostasis during fasting predicts protection from chemotherapy toxicity

Marta Barradas [1,13] ✉, Adrián Plaza [1,13] ✉, Gonzalo Colmenarejo [2], Iolanda Lázaro [3], Luis Filipe Costa-Machado[1], Roberto Martín-Hernández[2], Victor Micó[4], José Luis López-Aceituno[1], Jesús Herranz[2], Cristina Pantoja[1], Hector Tejero[5], Alberto Diaz-Ruiz [6], Fatima Al-Shahrour [5], Lidia Daimiel [4], Viviana Loria-Kohen [7], Ana Ramirez de Molina [7,8], Alejo Efeyan [9], Manuel Serrano [10], Oscar J. Pozo [11], Aleix Sala-Vila [3,12] & Pablo J. Fernandez-Marcos [1] ✉

Fasting exerts beneficial effects in mice and humans, including protection from chemotherapy toxicity. To explore the involved mechanisms, we collect blood from humans and mice before and after 36 or 24 hours of fasting, respectively, and measure lipid composition of erythrocyte membranes, circulating micro RNAs (miRNAs), and RNA expression at peripheral blood mononuclear cells (PBMCs). Fasting coordinately affects the proportion of polyunsaturated versus saturated and monounsaturated fatty acids at the erythrocyte membrane; and reduces the expression of insulin signaling-related genes in PBMCs. When fasted for 24 hours before and 24 hours after administration of oxaliplatin or doxorubicin, mice show a strong protection from toxicity in several tissues. Erythrocyte membrane lipids and PBMC gene expression define two separate groups of individuals that accurately predict a differential protection from chemotherapy toxicity, with important clinical implications. Our results reveal a mechanism of fasting associated with lipid homeostasis, and provide biomarkers of fasting to predict fasting-mediated protection from chemotherapy toxicity.

Fasting is a nutritional intervention consisting in the restriction of nutrient intake during a relatively long period of time. It elicits a profound metabolic reprogramming aimed at shifting nutrient supply from external food intake to internally stored nutrients. In mammals, the response to short-term fasting (from 12 to 48 h) in terms of nutrient mobilization through the bloodstream has been extensively studied[1,2]. It follows sequential phases, during which nutrients are released from different storing depots. First, glucose is released from glycogen stores in the liver and muscle. Upon depletion of glycogen, two fasting mechanisms are activated: adipose tissue activates lipolysis and releases free fatty acids (FFAs) into the bloodstream, reaching the liver where they are used to produce ketone bodies, a process termed ketogenesis. Also, gluconeogenesis is activated in the liver, generating

glucose mainly from glycerol (released during lipolysis), and from amino acids (originated mainly from autophagy and muscle breakdown). All these physiological responses are tightly regulated by hormonal and molecular mechanisms. At the hormonal level, fasting decreases blood insulin and leptin, and increases blood ghrelin and glucagon[3,4], while blood adiponectin remains unchanged[5]. Several signal transduction pathways are affected by fasting; one of them is the peroxisome proliferator-activated receptor α (PPARα), a nuclear receptor of fatty acids. PPARα becomes activated by the fasting-mediated increase in blood FFAs and triggers the expression of many target genes to coordinate the physiological switch from glucose to lipid catabolism in several tissues, including blood cells[6]. Also, sterol-regulatory element-binding protein (SREBP), a transcription factor

regulated by insulin and involved in the synthesis and modification of lipids (sterols and fatty acids) becomes repressed with fasting, and activated with refeeding in liver, adipose tissue and skeletal muscle[7].

Periodic activation of this complex response, termed periodic or intermittent fasting, elicits numerous protective effects against aging, metabolic alterations, neurological disorders and cardiovascular health[8–14]. Short-term fasting is also protective in different stress scenarios, including ischemia-reperfusion injury[15] or bouts of inflammation[16]. In addition, short-term fasting has been shown to reduce chemotherapy-induced toxicity in mice and humans[17–26] and to improve the anti-tumor efficacy of chemotherapy[27–30]. In mice, fasting for 48 or 60 h before etoposide treatment[26] or for 72 h before irinotecan treatment[20] strongly protected from chemotherapy toxicity, although these studies did not provide mechanistic explanations. In other studies, fasting mice for 24 h before etoposide chemotherapy showed enhanced survival of small intestinal epithelial stem cell by enhancing the DNA damage response and improving DNA protection by an undescribed mechanism[17]. Also, mice fasted for 72 h showed a strong decrease in blood insulin-like growth factor (IGF-1); and mice with a genetic modification that reduced the expression of IGF-1 in the liver were protected from toxicity of doxorubicin chemotherapy[24]. In humans, short-term fasting was first shown to improve toxicity parameters in a case series report with 10 patients with different types of tumors and chemotherapies, and with fasting times that ranged from 48 to 140 h prior to and 5 to 56 h after chemotherapy[18]. Subsequent studies confirmed this initial report: women with HER2-negative breast tumors that fasted for 24 h before and 24 h after docetaxel/doxorubicin/cyclophosphamide treatment showed increased erythrocyte and thrombocyte counts (indicating decreased bone marrow toxicity), and a protection from DNA damage in CD45+ CD3− myeloid cells[22]. A second study described patients receiving platinum-based chemotherapy and following different fasting times: one group fasting for 24 h pre-chemotherapy; another fasting for 48 h pre-chemotherapy; and another fasting for 72 h (48 h pre-chemotherapy and 24 h post-chemotherapy). This study reported decreased DNA damage (COMET assay) in peripheral blood mononuclear cells (PBMCs) only in patients that had fasted ≥48 h, but not in 24 h-fasting patients. These patients experienced decreased toxicity only in the groups with elevated ketone bodies. However, this protection did not correlate with IGF-1 decrease, that was strongest in the 24 h-fasting group that showed no improvement with fasting[21]. In an individually-randomized cross-over trial patients with gynecological cancer, a fasting period lasting from 36 h before chemotherapy to 24 h after the end of the chemotherapy significantly improved self-reported quality of life[23]. Another report showed that taking a fasting mimicking diet (FMD) for 3 days prior and following chemotherapy eliminated the need for dexamethasone and improved the frequency of radiologically complete or partial response[25]. Finally, a recent report showed that taking a FMD around chemotherapy administration improved the anti-tumor immune response and the general metabolic status of patients of different cancer types[30]. With all this in mind, it is not yet clear which of the multiple responses to short-term fasting is responsible for these beneficial effects.

In the current study, we investigate molecular mechanisms of short-term fasting in humans and mice, establishing the similarities and divergences between these two species. For this, we analyzed blood samples from healthy human volunteers subjected to 36 h of fasting or mice fasted for 24 h and measured erythrocyte membrane composition, gene expression in PBMCs and cell-free circulating miRNAs. We observed a coordinated evolution of all these parameters with fasting in both humans and mice, with the exception of circulating miRNAs that were not conserved in mice. We observed that the biomarkers that we establish helped to stratify individuals according to their different responses to fasting. Importantly, mice fasting during administration of chemotherapy were more protected from

**Table 1 | Morphometric and basic cardiovascular data from recruited volunteers**

|  | Mean ± SD | Range |
|---|---|---|
| **Female/male** | 10/10 |  |
| **Age (years)** | 27.1 ± 7.1 | 21–46 |
| **Height (cm)** | 172.1 ± 2 | 156.1–188.4 |
| **Body weight (kg)** | 70.0 ± 10.6 | 48.5–85 |
| **BMI (kg/m²)** | 23.6 ± 2.6 | 19.1–28.1 |
| **Obesity status** | 3 Overweight I<br>2 Overweight II<br>15 Normoweight |  |
| **Fat mass (%)** | 27.4 ± 10.2 | 12.6–44.3 |
| **Fat mass class** | 7 Elevated<br>13 Normal |  |
| **Muscle mass (%)** | 33.2 ± 7.2 | 23.7–44.1 |
| **Muscle mass class** | 5.0 ± 2.1 | 2–9 |
| **Basal metab. (kcal/day).** | 1531.1 ± 203.4 | 1173–1812 |
| **Waist circumf. (cm)** | 80.3 ± 7.7 | 66.4–95 |
| **Waist class** | 3 Risk<br>17 No Risk |  |
| **Systolic pressure (mm Hg)** | 122.8 ± 12.0 | 101–153 |
| **Systolic pressure class** | 1 High<br>4 Normal-High<br>15 Normal |  |
| **Diastolic pressure (mm Hg)** | 73.4 ± 10.3 | 59–95 |
| **Diastolic pressure class** | 3 High<br>17 Normal |  |
| **Heart rate (bpm)** | 66.7 ± 3 | 45–88 |

chemotherapy toxicity, and this protection strongly correlated with changes in their erythrocyte membrane fatty acids and gene expression at PBMCs. These parameters allowed to define two clear groups of individuals that were differentially protected from chemotherapy toxicity. We propose that our findings help explain why individuals with similar biochemical parameters display different global responses to fasting and different levels of protection from chemotherapy toxicity by fasting. Also, our results could serve as a starting point to the development of personalized nutritional, behavioral or pharmacological strategies to improve fasting-mediated protection from chemotherapy toxicity.

## Results

### Standard biomarkers of fasting in healthy humans and mice

We recruited a study sample of 20 human volunteers. Baseline morphometric, physiological and metabolic data included in the trial are shown in Table 1. The cohort had an equal number of males and females, from 21 to 46 years of age, and 25% of the individuals presented overweight (body-mass index BMI >25 kg/m²), which is representative of the average Spanish population[31]. Participants were subjected to a short-term fasting protocol of 36 h, a fasting time similar to previous reports[6,32–34], and then to 24 h of refeeding, as depicted in Fig. 1a. We obtained blood samples from three time points: basal (after 12 h of fasting), fasting (after 36 h of fasting), and refed (after 12 h of refeeding and 12 h of fasting).

To compare the response to fasting between human and mouse, we subjected 4 male C57BL/6JOlaHsd mice of 12 weeks of age to fasting for 48 h and measured their blood levels of glucose and the ketone body β-hydroxybutyrate (β-HB), as well as their body weight, at different times. According to the literature, human volunteers reach their maximum blood β-HB levels (~3–4 mM) after 72 h of fasting, presenting ~1 mM after 24 h of fasting, and ~2 mM after 48 h of fasting[33]. We measured the evolution of blood β-HB with time in mice and observed that ketone bodies were similar to those observed in humans: ~1 mM

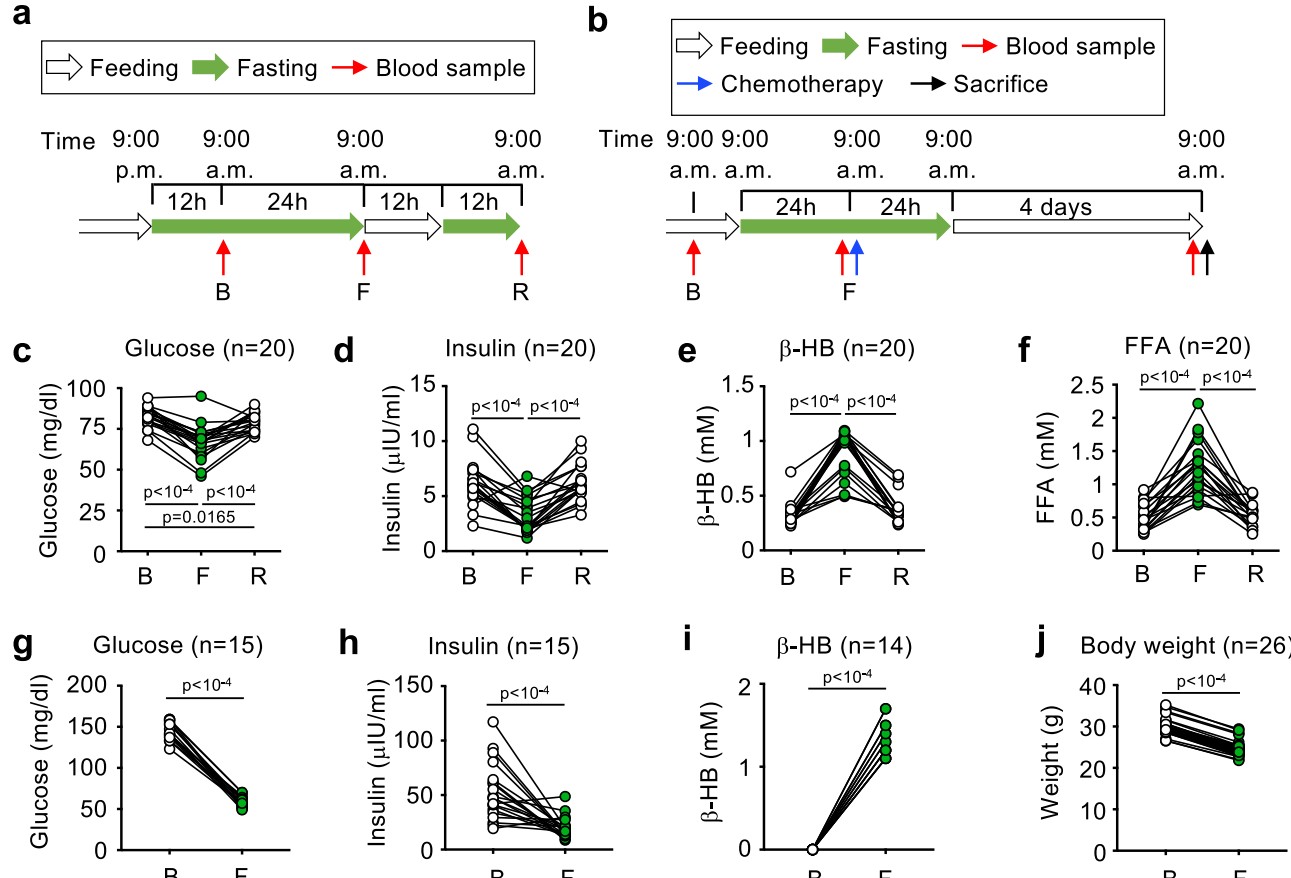

**Fig. 1 | Standard biomarkers of short-term fasting in healthy humans and mice. a**, **b** Scheme of the protocols followed for the short-term fasting experiment in humans (**a**) and mice (**b**). Green arrows indicate the times of fasting, and white arrows indicate normal feeding. Red arrows indicate blood sampling; blue arrows indicate chemotherapy administration, and black arrows indicate sacrifice. **c–f** Blood levels in human volunteers of glucose (**c**), insulin (**d**), β-hydroxybutyrate (β-HB, **e**) and free fatty acids (FFAs, **f**) at the basal point after 12 h of overnight fasting (B); after 36 h of fasting (F); and after 12 h of refeeding and 12 h of overnight fasting (R). **g–i** Blood levels of glucose (**g**), insulin (**h**) and β-HB (**i**) in fed conditions at first time in the morning (Basal, B) and after 24 h of fasting (F). **j** Mouse body weight at the indicated times. Each dot represents the data from a single individual, and is linked with a straight black line with the data of the same individual in the next point of the time course experiment. Statistical significance was performed using the one-way ANOVA test with Tukey correction for multiple comparisons (**c–f**); or the paired two-tailed Student *t* test (**g–j**). The exact *p* value is provided for each significant comparison. Source data are provided as a Source Data file.

after 24 h, and ~2 mM after 48 h of fasting (Fig. S1a). Next, we measured glucose evolution with fasting, and we observed that glucose went down until 50 mg/dl after 48 h (Fig. S1b), a similar value to that obtained in fasting humans for 36 h (Fig. 1c). Finally, according to previous reports[33], human volunteers lost ~1 kg after 24 h of fasting, and up to ~4 kg after 72 h of fasting; considering the average body weight of volunteers in that study, this body weight loss represented 1.3% of the total body weight after 24 h fasting, and up to 5.2% after 72 h of fasting. In contrast, mice fasted for 24 h lost 13.5% of their initial body weight; and 18.8% after 48 h of fasting (Fig. S1c). Based on these biochemical and morphometric data, we decided to subject a new cohort of 26 male mice to 24 h of fasting, a time at which biochemical parameters were very similar between mice and humans, and gathered morphometric data and blood samples before and after this fasting period. These same mice were then treated with chemotherapy to evaluate their chemotherapy toxicity, as will be explained later (for a complete scheme, see Fig. 1b).

In humans, plasma samples from three time points (basal, 36 h fasting and 24 h post-refeeding) were analyzed for several biochemical parameters: glucose, insulin, ketone bodies (β-HB), FFAs, succinate and leptin. These blood metabolites have been well studied in humans during similar fasting periods[6,32–35]. As expected, blood glucose, insulin and leptin decreased with fasting (Fig. 1c, d, Fig. S1d), while β-HB and FFAs increased (Fig. 1e, f). Basal levels of these parameters were

recovered after the refeeding period (Fig. 1c–f, Fig. S1d). We also observed a significant increase in the blood levels of succinate after 36 h of fasting, confirming previous reports[36] (Fig. S1e). In our new cohort of 26 mice, we observed again that fasting decreased blood glucose and insulin (Fig. 1g, h), increased blood β-HB (Fig. 1i) and decreased body weight (Fig. 1j).

**Coordinated changes in membrane fatty acids with short-term fasting**

Fasting results in a dramatic change in nutrient usage, shifting from glucose to fatty acids that are released by the adipose depots into the bloodstream and used by the liver to produce ketone bodies. We hypothesized that this drastic change in blood fatty acids could have an impact on the lipid composition of cellular membranes, an unexplored effect of fasting. We quantified erythrocyte membrane fatty acids by gas chromatography in the blood samples from our human and mouse cohorts. Significantly changed FFAs are represented in Fig. 2a, b, and all the analyzed FFAs in Fig. S2a, b. Changes in the fatty acid profile of cell membranes with nutritional interventions in humans usually require several days to become apparent; in accordance, most membrane lipid species did not change with fasting in humans (Fig. 2a, Fig. S2a). The only fatty acid that changed significantly at the fasting point was C18:2n6. Most significant changes in human membrane fatty acids only appeared at the refeeding state, which

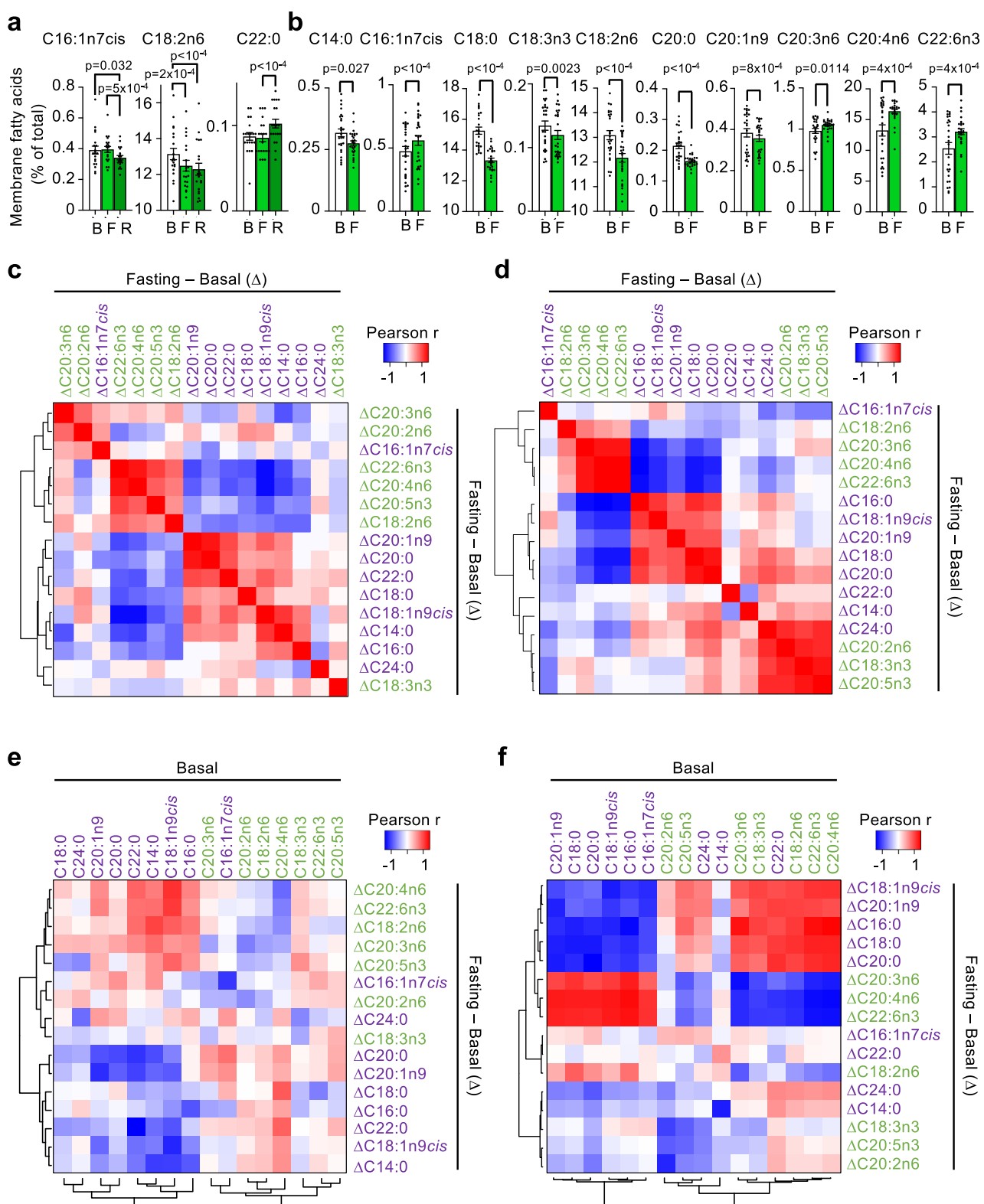

**Fig. 2 | Short-term fasting elicits changes in erythrocyte membrane fatty acids.** **a**, **b** Levels of the indicated fatty acids in erythrocyte membranes isolated at the indicated times of the fasting experiments (B: basal, after 12 h of fasting; F: after 36 h of fasting; R: after 12 h of refeeding and 12 h of fasting) in humans (**a**, *n* = 20) and mice (**b**, *n* = 26), expressed as % of total membrane fatty acids. **c**, **d** Heatmaps depicting the Pearson correlation r coefficients between changes (fasting *minus* basal, Δ) of all the analyzed erythrocyte membrane fatty acids from human (**c**) or mice (**d**) samples. Red color indicates direct correlations, and blue color indicates indirect correlations. Saturated and monounsaturated fatty acids are written in purple; polyunsaturated fatty acids are written in green. The precise order of each fatty acid was determined using a cluster analysis. **e**, **f** Heatmaps representing the Pearson correlation r coefficients between the changes (fasting *minus* basal) and the basal levels of the same erythrocyte membrane fatty acids shown in **a**, **b**. Bars represent the average of the indicated number of individuals. Error bars indicate the standard error of the mean. Statistical significance was assayed using the one-way ANOVA test with Tukey correction for multiple comparisons (**a**), the unpaired two-tailed Student *t* test (**b**); or the Pearson correlation test (**c–f**). The exact *p* value is provided for each significant comparison. Source data are provided as a Source Data file.

given the long lifespan of erythrocytes might reflect changes initiated during fasting. In contrast, we observed that mice showed more intense changes in erythrocyte membrane fatty acids than humans: 24 h of fasting in mice already induced significant changes in several membrane fatty acids, including a significant reduction in C18:2n6, in coincidence with humans (Fig. 2b, Fig. S2b).

Next, we explored if the changes in membrane fatty acids in erythrocytes were also observed in other tissues. Changes in lipid composition of erythrocyte membranes have been shown to reflect changes in the lipid composition of cellular membranes of other tissues with different conditions, such as cardiomyocytes upon cardiac transplantation[37]; or liver and kidney upon liver disease[38], but this has not been specifically studied with fasting yet. We measured the membrane fatty acids of erythrocytes, liver, heart, kidney and brain samples obtained from fed control or 24 h fasted mice. As a control of fasting, fasted mice showed decreased blood glucose and increased blood ketone bodies compared with fed control mice (Fig. S2c). Erythrocytes from fasted mice changed each of their membrane fatty acids in the same direction than presented in Fig. S2b, although only a few of them (α-linolenic acid C18:3n3, linoleic acid C18:2n6 and docosahexaenoic acid C22:6n3) reached significance (Fig. S2d). This difference in statistical significance was expected, since fed and fasted samples are not paired in this experiment, and the number of individuals is much lower. Once we validated the reproducibility of the changes in erythrocyte fatty acids, we measured membrane fatty acids in the other collected tissues. We observed a marked lipid response in the liver, where almost all minority fatty acids were decreased with fasting, and there was a significant increase in the abundant fatty acid C22:6n3 (docosahexaenoic acid), reflecting the strong role of the liver in lipid management during fasting (Fig. S2e). In the other tissues these changes were less evident. In kidney, only C18:3n3, C20:3n6 (cis-8,1,14-eicosatrienoic acid) and C18:2n6 changed significantly with fasting, following the opposite direction than in erythrocytes (Fig. S2f). In heart, we observed many fatty acids changed, especially minor fatty acids; and all of them followed the same direction than in erythrocytes, with the exception of C20:3n6, that was decreased upon fasting (Fig. S2g). Finally, fasting induced an almost significant change in C18:3n3, C18:2n6, C14:0 (myristic acid) and C18:1n9cis (oleic acid) in brain (Fig. S2h). In all, these results indicate that fasting induces changes in the fatty acid composition of the membrane of different organs, and that this response is not identical between the organs studied. Since blood sampling is not invasive, and it is clinically relevant as a convenient source of biomarkers, we focused our attention on erythrocytes for the following experiments.

We then studied in detail the evolution of membrane fatty acids with fasting. For this, we calculated for each membrane fatty acid the difference between fasting and basal time points (represented by Δ values in the figures). Then, we correlated changes in each membrane fatty acid. Two interesting findings came from this analysis: first, changes in most human polyunsaturated membrane fatty acids (ΔPUFAs) correlated directly among them (Fig. 2c, names of PUFAs marked in green). This finding revealed a coordinated evolution of these types of membrane fatty acids during short-term fasting. In mice we observed similar results: one group of fatty acids composed mainly of PUFAs (C18:2n6, C20:3n6, C20:4n6 and C22:6n3) changed coordinately between them (Fig. 2d). The second relevant finding was that ΔPUFAs correlated inversely to changes in most monounsaturated (ΔMUFAs) and saturated fatty acids (ΔSFAs) (Fig. 2c, d, names of MUFAs and SFAs marked in purple), that in turn evolved in coordination among themselves. From these findings, we concluded that there were two groups of membrane fatty acids (PUFAs vs. SFAs and MUFAs) that evolved inversely with fasting. In mice, where net changes in membrane fatty acids were more intense than in humans, we observed that most fatty acids that were significantly altered with fasting showed very strong correlations, including the SFAs and MUFAs C14:0, C18:0,

C20:0 or C20:1n9, all of which decreased with fasting; and the PUFAs C20:3n6, C20:4n6 and C22:6n3, all of which increased with fasting (Fig. 2d). Importantly, these last three PUFAs were, precisely, the ones that evolved opposite to SFAs and MUFAs with fasting, while the three PUFAs that clustered closer to SFAs and MUFAs did not change or decreased with fasting (Fig. S2b). We will focus on the PUFAs that increased with fasting and evolved in an opposite direction to SFAs and MUFAs for the following analysis.

We wondered if this coordinated evolution of fatty acids with fasting was associated with the basal metabolic status at the time of fasting. To test this hypothesis, we calculated the Pearson correlation r coefficients between the changes of each fatty acid with fasting, and the basal status of each fatty acid. As shown in Fig. 2e, f, there was a clear correlation in both humans and mice between changes with fasting and basal levels of each fatty acid. Importantly, these correlations were not stochastic but, instead, they again grouped fatty acids in the same two groups of PUFAs vs. MUFAs and SFAs. Remarkably, the PUFAs that correlated most clearly in mouse blood in Fig. 2f were C20:3n6, C20:4n6 and C22:6n3, exactly the same three PUFAs that increased with fasting (Fig. 2b) and whose changes with fasting correlated inversely with the group of SFAs and MUFAs (Fig. 2d). These results indicated that individuals with lower PUFAs at the basal point, were the ones that increased these PUFAs more strongly with fasting. In turn, those that, at the basal point, had higher SFAs and MUFAs, were the ones that decreased more strongly their SFAs and MUFAs with fasting.

The fatty acid composition of the plasma membrane is a dynamic result of many cellular processes, including lipid biosynthesis and enzymatic activities. Therefore, we also wanted to explore the activity of enzymes modulating or being affected by membrane lipid composition. For example, neutral sphingomyelinase (nSMase) has been shown to be activated by accumulated plasma membrane bending energy, strongly influenced by lipid composition[39]; and phospholipase A2 (PLA2) has been shown to increase membrane bending energy and activate neutral sphingomyelinase, which in turn can activate phospholipase A2[40]. We observed a significant increase in the activities of nSMase and cytosolic phospholipase (cPLA2) after 24 h of fasting in livers from fasted mice, compared to livers from fed controls (Fig. S2i, j).

## Changes in plasma circulating miRNAs during short-term fasting

Circulating miRNAs have become a very powerful source of biomarkers for different health situations. Previous reports studied serum circulating miRNAs in humans after a short, overnight fasting, and found no differences[41]. Therefore, we wanted to establish the dynamics of circulating miRNAs during a longer, 36 h fasting, and to compare them between humans and mice. We first analyzed a panel of 179 circulating miRNAs in the cell-free plasma samples from a subset of 4 human volunteers at basal and fasting points. From this panel, we identified 8 miRNAs with evident changes between the basal and fasting points, and validated them with the plasma samples from the volunteers with enough amount of plasma. This way, we confirmed changes with fasting and/or refeeding in 5 circulating miRNAs (Fig. 3a): decreased levels of miR-29b-3p (0.69-fold over basal), miR-194-5p (0.45-fold over basal), miR-141-3p (0.50-fold over basal) and miR-100-5p (0.76-fold over basal); and increased levels of miR-485-3p (1.34-fold over basal). We analyzed the predicted target genes from these altered plasma miRNAs, and clustered them according to their KEGG function (Fig. 3b). Interestingly, the predicted targets of miR-29b-3p and miR-100-5p showed the most robust associations with multiple KEGG functions, and particularly with those related to lipid metabolism: fatty acid biosynthesis, fatty acid metabolism and glycosphingolipid biosynthesis. We next analyzed miR-29b-30, miR-100-5p, miR-141-3p, and miR-194-5p in plasma samples from a subset of our fasted mice. Since miR-485-3p was not conserved between mice and humans, we could

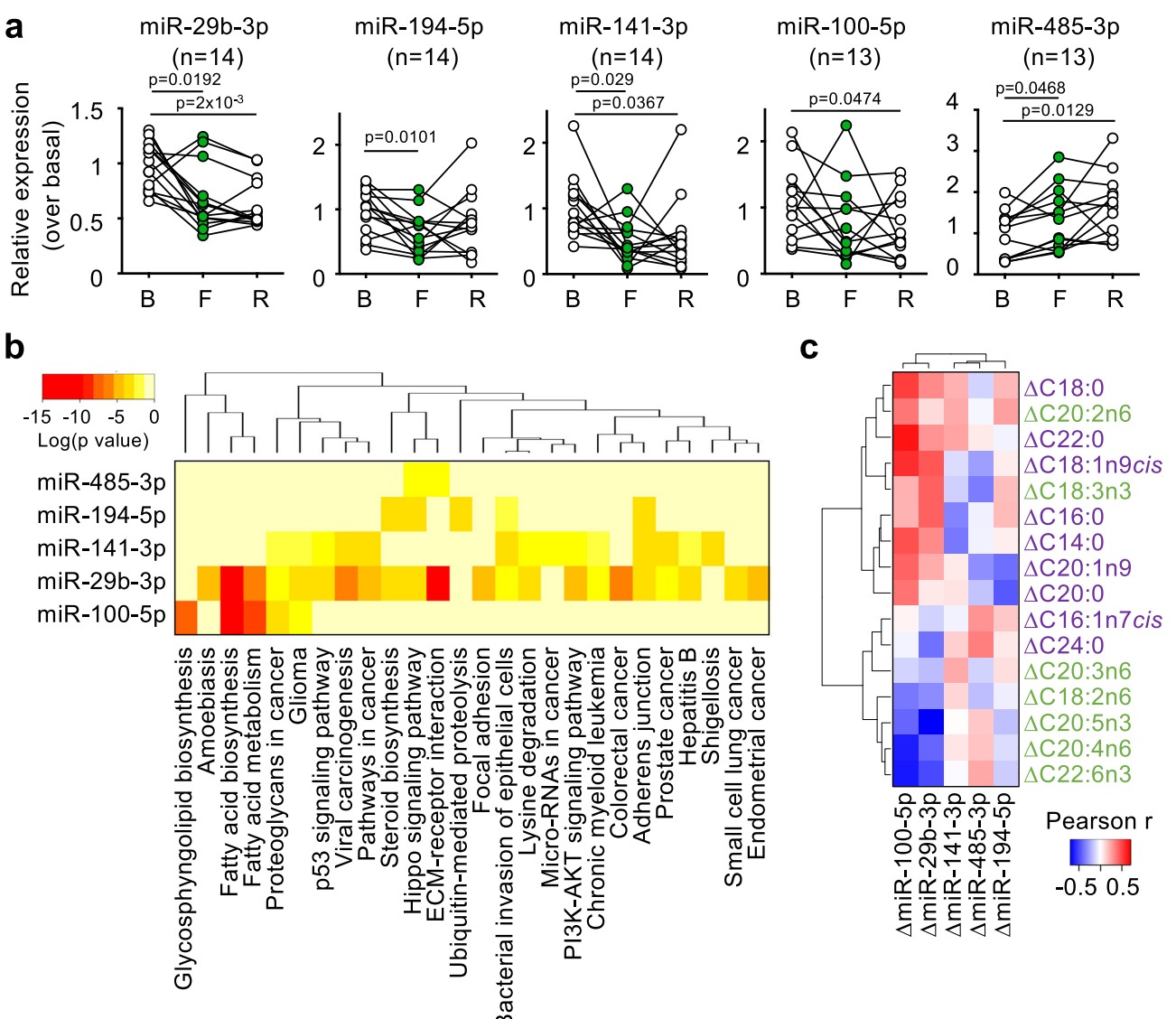

**Fig. 3 | Human circulating miRNAs change during short-term fasting.**
**a** Expression of the indicated miRNAs measured in plasma from human volunteers at basal (B), fasting (F) and refeeding (R) points. **b** Heatmap showing the KEGG functions associated with genes targeted by the indicated miRNAs. **c** Heatmap showing the Pearson correlation r values between changes (fasting *minus* basal, Δ) of all the analyzed erythrocyte membrane fatty acids and changes with fasting of the indicated circulating miRNAs significantly altered with fasting. Each dot represents the data from a single individual, and is linked with a straight black line with the data of the same individual in the next point of the time course experiment. Statistical significance was performed using the one-way ANOVA test with Tukey correction for multiple comparisons (**a**) or with Pearson correlations (**c**). The exact *p* value is provided for each significant comparison. Source data are provided as a Source Data file.

not analyze it in our serum samples. We did not observe any significant changes in these circulating miRNAs (Fig. S3a). These results suggest that changes in these circulating miRNAs with fasting are not conserved between humans and mice.

We then explored possible associations with fasting between human erythrocyte membrane fatty acids and circulating miRNAs. Remarkably, changes in human circulating miRNAs during fasting very efficiently distinguished between two groups of membrane fatty acids, almost identical to those defined in the correlation plot from Fig. 2c: a first group mostly comprising PUFAs and a second group mostly comprising SFAs and MUFAs (Fig. 3c). Of note, the two miRNAs whose changes most strongly defined these two groups of human erythrocyte membrane fatty acids were miR-29b-3p and miR-100-5p (Fig. 3c), whose target genes were the most strongly associated to lipid metabolism (Fig. 3b). As examples, see Fig. S3b, showing the inverse correlations of ΔmiR-100-5p with ΔC22:6n3, and the direct correlation of ΔmiR-100-5p with ΔC22:0, respectively. In contrast, when we

calculated the Pearson correlations between changes with fasting of these miRNAs in mouse plasma and erythrocyte membrane fatty acids, the mouse plasma miRNAs did not separate erythrocyte membrane fatty acids in the same two groups (PUFAs vs. SFAs and MUFAs, Fig. S3c), corroborating that circulating miRNAs did not evolve in the same way between humans and mice.

### Regulation of gene expression with fasting in human and mouse PBMCs

We wanted to explore changes in gene expression in PBMCs, given the utility of this type of samples to monitor the fasting response. For this, we purified PBMCs from blood samples of human volunteers and mice, and isolated their total RNA. In humans, we first confirmed the induction with fasting in PBMCs of the expression of PPARα target genes *PLIN2, CPT1A* and *SLC25A20*, as previously described[6] (Fig. S4a). However, in mouse PBMCs these genes were not clearly upregulated with fasting and, in the case of *Plin2*, fasting actually reduced its

expression (Fig. S4b) indicating that PPARα target genes are regulated differently with fasting in mouse and human PBMCs.

Next, based on the remarkable changes in membrane fatty acids (Fig. 2) and in circulating miRNAs in humans regulating fatty acid metabolism (Fig. 3), we studied one of the main regulators of fatty acid biosynthesis: the transcription factor sterol-regulatory element-binding protein 1 (SREBP-1, codified by the *SREBF1* gene), its regulator insulin-induced gene 1 (*INSIG1*) and its target stearoyl Co-A desaturase (*SCD*), that have been previously studied during fasting and refeeding in different tissues, including liver ([7,42] and Fig. S4c, top panel) and PBMCs ([43] and Fig. S4c, bottom panel). Human PBMCs showed a significant reduction in the expression of *SREBF1* (Fig. 4a). We also identified *INSIG1* as a gene targeted by both circulating miRNAs miR-29b-3p and miR-100-5p (Table 2) and by SREBP-1 itself[44], associated to lipid metabolism and SREBP-1 function[45]. We observed a decreased mRNA expression with fasting of *INSIG1* in human PBMCs (Fig. 4a). In turn, expression of the SREBP-1 target gene *SCD* did not change in a significant manner with fasting in human PBMCs with fasting, but was induced with refeeding (Fig. 4a). In mice, PBMCs also showed reduced mRNA expression of the *Srebf1* gene, its regulator *Insig1* and its target *Scd1* with fasting (Fig. 4b). These data indicated that short-term fasting elicits a coordinated mRNA expression reprogramming of fatty acid metabolism that also affects blood cells in humans and mice.

After confirming that the RNA from human PBMCs responded to our nutritional intervention, since they showed the previously described increased levels of PPARα target genes[6] (Fig. S4a) and decreased levels of SREBP pathway genes (Fig. 4a), we performed RNAseq with our human PBMC samples, and looked for genes whose changes with fasting correlated with changes in the fasting lipid response biomarkers (erythrocyte membrane fatty acids and circulating miRNAs). For this, we first selected all genes detected at the RNAseq related to the term "Lipid metabolic process" (GO:0006629, 923 genes in total). As a first approach, we calculated the Pearson correlation coefficients of the changes with fasting in all lipid-related genes, with changes in membrane fatty acids. We observed a remarkable ability of this set of genes to segregate changes in membrane fatty acids in two groups of ΔPUFAs vs. ΔSFAs and ΔMUFAs (Fig. S4d compared with Figs. 2c, 3c). This result suggests that within this large number of lipid-associated genes, many of them are associated with the changes in membrane fatty acids during fasting. From these 923 lipid-related genes, we selected the 10 genes with the highest average Pearson correlation coefficients with membrane fatty acids and lipid-associated circulating miRNAs. As expected, these genes clearly segregated ΔPUFAs vs. ΔSFAs and ΔMUFAs, and strongly correlated with ΔmiR-29b-3p and ΔmiR-100-5p (Fig. 4c). Most remarkably, when we analyzed the functions of this set of mRNA biomarkers, we noticed that a high proportion of them (8 out of 10) were associated with the insulin-PI3K-AKT signaling pathway: *TTC7B*[46], *THEM4*[47], *TRIB3*[48], *MTMR7*[49], *VAV3*[50], *ACAD10*[51], *LPCAT1*[52], and *INPP5A*[53]. We analyzed this list of genes in the PBMCs isolated from mice. Although many of them were not detectable in our mouse PBMC samples, we could quantify 4 of these genes (*Lpcat*, *Erg28*, *Vav3*, and *Inpp5a*). As in humans, these mouse genes strongly segregated the erythrocyte membrane fatty acids in two separate groups of ΔPUFAs vs. ΔSFAs and ΔMUFAs (Fig. 4d). Of note, when we analyzed the changes with fasting in fed and fasted mouse livers and PBMCs of the SREBP-1 pathway genes and the genes identified in our RNAseq experiment, we observed that changes with fasting of the expression of the SREBP-1 pathway genes *Srebf1*, *Scd1*, and *Insig1* coincided between liver and PBMCs (Fig. S4c). In the case of the genes identified in our RNAseq experiment, only changes with fasting of the expression of *Erg28* coincided between liver and PBMCs; while *Lpcat*, *Vav3*, and *Inpp5a* increased with fasting in liver but decreased or were unchanged in PBMCs (Fig. S4c). In all, the evolution with fasting of this set of genes in PBMCs suggests a close relationship between

insulin signaling, membrane fatty acids and circulating miRNAs during fasting, and defines potential biomarkers of this response.

We wondered if changes in the parameters of fasting that we identified (erythrocyte membrane fatty acids, lipid-associated circulating miRNAs miR-29b-3p and miR-100-5p, and the lipid-associated mRNA biomarkers) correlated with other biochemical and morphometric parameters that we gathered from our experiments, detailed in Supplementary Table 1. As shown in Table 3, the two biochemical parameters in humans that associated more strongly with our fasting biomarkers were basal Homeostatic Model Assessment for Insulin Resistance (HOMA-IR) and basal insulinemia, obtained after 12 h of fasting (see Fig. 1a). In parallel, the two most robust biochemical associations with the fasting parameters in mice were fasting HOMA-IR and fasting insulinemia, obtained after 24 h of fasting. As a graphical description, Pearson correlations between changes with fasting of membrane fatty acids and the most significant biochemical and morphometric parameters are represented in Fig. S4e (humans) and S4f (mice). In all, these results indicate that insulin sensitivity measured before and after a fasting period is a strong predictor of the fasting-triggered mechanisms described here, and strongly correlates in a coordinated manner with the evolution with fasting of the two groups of membrane fatty acids: ΔPUFAs vs. ΔMUFAs/ΔSFAs.

Finally, we wondered if all the fasting biomarkers that we identified could be associated with differential responses to fasting. If this was the case, we should be able to distinguish between two or more clearly different groups of individuals, according to their fasting biomarkers. In order to compare biomarkers with very different ranges of values, we calculated the median of each fasting parameter for all individuals. Then, we placed each individual as above or below this median, as shown in Fig. 4e (humans) and 4f (mice) (red squares mean above the median, and blue squares mean below the median). This way, two groups of individuals, that we called Group A and Group B, could be clearly identified according to their response to fasting in mice and humans.

## Fasting protects from oxaliplatin-induced multi-organ toxicity in mice

Fasting has been shown to protect from several insults, including chemotherapy toxicity[17–23]. Based on this, we decided to evaluate the biological relevance of these fasting biomarkers identified above, and the groups defined by them, in mice subjected to chemotherapy. First, we generated a new cohort of mice (*n* = 6), different from that used in Figs. 1–4, to set up a battery of toxicity biomarkers. Mice were inoculated with 15 mg/kg oxaliplatin, a common chemotherapy for intestinal, stomach and pancreatic tumors, and either fed ad libitum, or fasted for 24 h before and 24 h after chemotherapy administration (a total of 48 h). A few days before chemotherapy, a basal von Frey neurotoxicity score was obtained, and 4 days after chemotherapy, a new von Frey neurotoxicity score was measured. One day later, mice were sacrificed and tissues collected, including liver, kidney, heart, dorsal root ganglia, spleen and peripheral blood (for a scheme, see Fig. 1b).

The most common oxaliplatin side effects, according to the existing literature[54–58], are: neurotoxicity, the main responsible for discontinuation of oxaliplatin treatment, even more than tumor progression[54]; hematological toxicity, most commonly neutropenia (more than 1/3 of patients) and, more rarely, thrombocytopenia[57]; liver damage, including sinusoidal obstruction and steatohepatitis[55]; acute kidney damage[56]; increased cardiotoxicity[58]; and gastrointestinal toxicity in the form of easily treated conditions such as diarrhea, vomits, nausea or mucositis[59]. We analyzed a large series of toxicity-associated genes in kidney[60–62], heart[63–66], liver[67–75], nervous system (in dorsal root ganglia)[76,77], and blood cells[57]. Remarkably, fasting prevented the increase in all of the tested oxaliplatin-responding genes

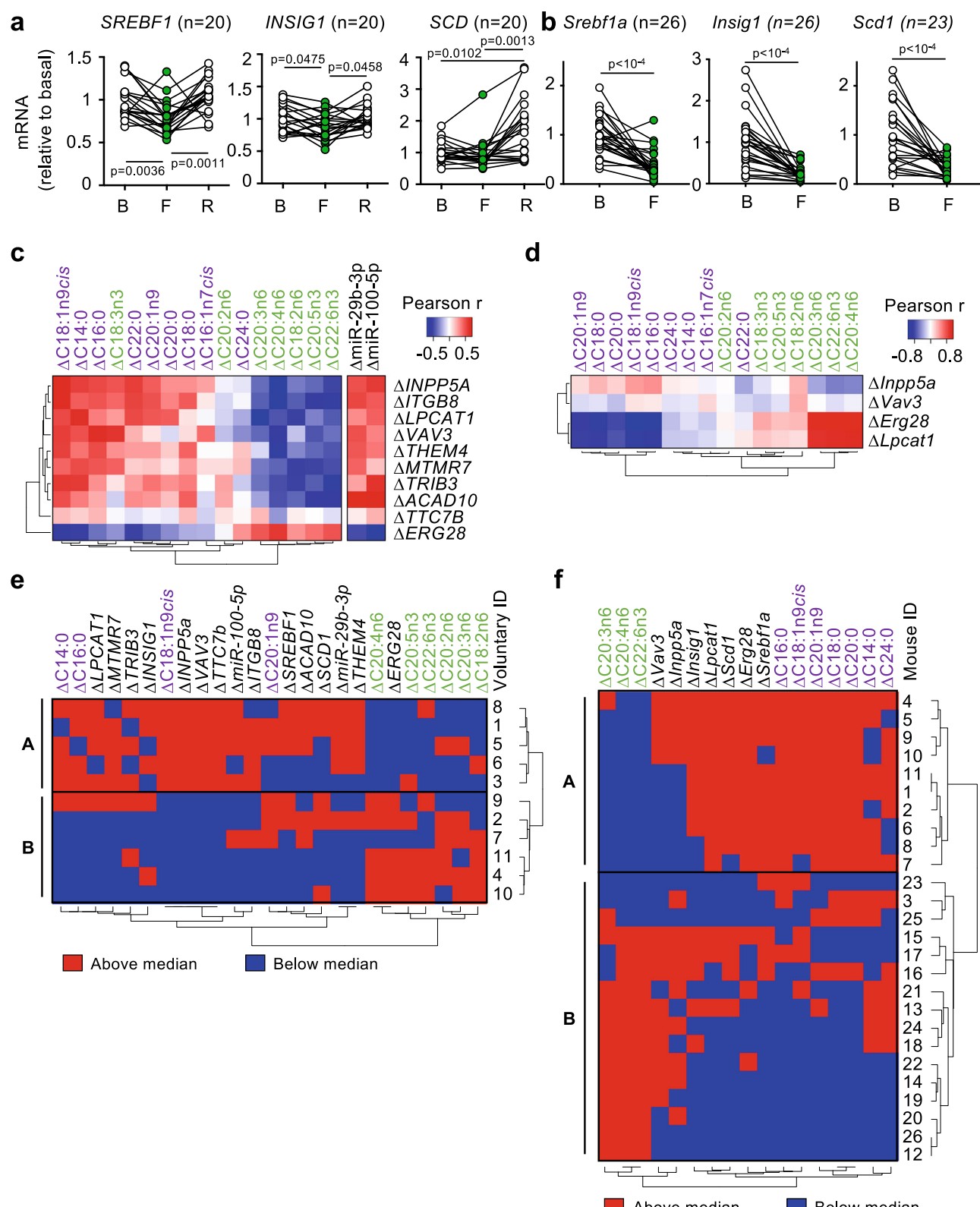

**Fig. 4 | Gene expression in peripheral blood mononucleated cells (PBMCs) during short-term fasting. a, b** Total RNA from PBMCs isolated from human volunteers (**a**) or mice (**b**) at the indicated time points (B basal, F fasting, R refeeding) was analyzed for the expression of the indicated lipid metabolism genes. **c, d** Heatmaps depicting the Pearson correlation r coefficients in humans (**c**) or mice (**d**) between the changes with fasting of erythrocyte membrane fatty acids and/or circulating miRNAs, and the changes in mRNA expression of the indicated genes identified in the PBMC RNAseq experiment. **e, f** Heatmaps depicting each individual, represented by a number next to each panel, and its position with respect to the median value for each of the indicated parameters: red indicates "above the median", and blue indicates "below the median". Both individuals and parameters were ordered according to their similarity. This unsupervised clustering defined two groups of individuals named "A" and "B" in the graphs. Line-connected dots represent the paired data for each individual (**a, b**). Statistical analysis was performed using the one-way ANOVA test with Tukey correction for multiple comparisons (**a**); the paired two-tailed Student $t$ test (**b**); or Pearson correlations (**c, d**). The exact $p$ value is provided for each significant comparison. Source data are provided as a Source Data file.

(Fig. 5a–d, Fig. S5a). Also, fasted mice were protected from oxaliplatin-induced increased paw sensitivity (quantified by the von Frey test in Fig. 5d, panel to the right), and showed higher blood platelet numbers 5 days after chemotherapy (Fig. 5e) than fed mice. In contrast, fasted mice did not show any improvement in their oxaliplatin-induced neutropenia (Fig. S5b). Histologically, we measured different pathological findings: in liver, we quantified hepatocyte polyploidy, sinusoidal dilatation and lipid vacuoles. In kidney, we quantified Bowman capsule dilatation glomerulopathy and tubule dilatation. From these histological analyses, we found a remarkable protection from oxaliplatin toxicity in both liver (Fig. 5f, h) and kidney (Fig. 5g, i). We did not detect significant histological differences in paraffin sections from the heart, where we analyzed cardiomyopathies as multifocal necrosis, small number of cardiomyocytes and myofiber vacuolation; hemorrhages and infiltration. As a result of all these protection parameters, fasted mice lost less body weight than fed mice (Fig. 5j).

We have also measured the effects of fasting in the toxicity elicited by doxorubicin, a chemotherapy unrelated to oxaliplatin used against lung, gastric, ovarian, breast, thyroid and pediatric cancers. The main doxorubicin secondary effect is cardiotoxicity[78], followed by leukopenia[79], hepatotoxicity[80] or renal toxicity[81,82]. We have measured toxicity in these tissues after an intraperitoneal inoculation of 15 mg/kg of doxorubicin in 12–14-week-old male mice, using the same fasting protocol and tissue collection than described for oxaliplatin (Fig. 1b). Fasting protected from heart (Fig. S5c), kidney (Fig. S5d), and liver (Fig. S5e) damages; and from leukopenia (most strongly monocyte counts, but also showing a clear trend in total white blood cells and lymphocyte counts, Fig. S5f), resulting in a much stronger recovery of body weight after chemotherapy administration, compared with their respective chemotherapy-treated fed controls (Fig. S5g).

Finally, we also studied the protective effects of combining a shorter bout of fasting (12 h before and 12 h after, amounting to 24 h in total) with 15 mg/kg oxaliplatin chemotherapy. As shown in Fig. S5h–m, 24 h of fasting prevented the increase in several biomarkers of toxicity, such as *Afp*, *Hmox1* (but not *Ccl2* or *Ctgf*) in liver (Fig. S5h); *Atf3* and *Anf1* (but not *Myh14*) in heart (Fig. S5i); and *Lcn2*, *Kim1* and *B2m* in kidney (Fig. S5j). 24 hours of fasting did not prevent oxaliplatin-induced thrombocytopenia (Fig. S5k); improved the chemotherapy-induced loss of body weight (Fig. S5l); and protected from some biomarkers of neurotoxicity, as expression of the *Myl1*, *Myh4*, *Tnnc2*, *Tnn3*, and *Cd74* genes, but did not protect from increased paw sensitivity in the von Frey assay (Fig. S5m).

## Lipid homeostasis biomarkers correlate with the fasting-mediated protection from chemotherapy toxicity

We next wondered if our fasting biomarkers could help predict the response to fasting between different individual mice. For this, the same 26 mice that were used to identify fasting biomarkers in Figs. 1–4 were treated with 15 mg/kg oxaliplatin with a fasting period of 24 h before and 24 h after chemotherapy (48 h of fasting in total, for a scheme see Fig. 1b). This large number of mice would allow us to study possible differential effects of fasting with enough individuals. To control for fasting efficacy, we also added one small group of 5 mice treated with 15 mg/kg oxaliplatin without fasting (fed control). First, we validated that, in this new cohort, fasting protected from oxaliplatin toxicity in all tested tissues (Fig. S6a). Next, we calculated the Pearson correlations between the fasting parameters and toxicity biomarkers. For fatty acid homeostasis we used the SFAs, MUFAs and PUFAs that showed the most clearly opposed evolution with fasting (Fig. 2d): the SFAs and MUFAs C14:0, C16:0, C18:0, C18:1n9cis, C20:0, C20:1n9, C24:0; and the PUFAs C20:3n6, C20:4n6, C22:6n3. We also included in this analysis the changes in expression in PBMCs of the lipid metabolism genes *Srebf1a*, *Insig1*, and *Scd1*, altered with fasting (Fig. 4b); and of the genes *Lpcat1*, *Erg28*, *Vav3* and *Inpp5a*, that we identified using the RNAseq experiment in human PBMCs (Fig. 4d; the rest of the genes identified in the RNAseq from Fig. 4d could not be detected in mouse PBMCs). As shown in Fig. 6a, there was a very strong Pearson correlation between the fasting and the toxicity biomarkers in all tissues analyzed, including kidney (*Tgf-b*, *Il1-b*, *Lcn2*), liver (*Ctgf*, *Tgf-b*, *Il1-b*, *Cyp2e1*, *Sult1c2*, *Ccl2*, *Casp3*, *Afp*, *Hmox1*), heart (*Anf1*, *Tgf-b*, *Il1-b*, *Myh14*, *Atf3*) and bone marrow (erythrocyte count, hemoglobin, hematocrit and platelet count). Of note, kidney expression of the *Lcn2* toxicity marker behaved in the opposite direction compared with all the other markers. Higher tissue weight after chemotherapy reflects a lower damage by chemotherapy in these tissues, and we observed a

## Table 2 | List of the genes targeted by both miR-29b-3p and miR-100-5p in humans

| List of the genes targeted by both miR-100-5p and miR-29b-3p in humans |
| --- |
| *INSIG1* |
| *YWHAE* |
| *MORF4L2* |
| *SET* |
| *PRRC2C* |
| *ESR1* |
| *CCNYL1* |
| *DNMT1* |
| *C21orf91* |
| *COL4A1* |
| *IMPDH1* |
| *YBX3* |
| *PTP4A1* |
| *CTC1* |
| *NFIA* |

## Table 3 | Absolute value of the average Pearson correlation r coefficients of the indicated parameters with the most robust biomarkers from humans and mice shown in Fig. 4c, d

| Human | |
| --- | --- |
| **Parameter** | **Average Pearson r** |
| Basal HOMA-IR | 0.365 |
| Basal insulin | 0.343 |
| Fasting β-HB | 0.283 |
| Basal β-HB | 0.271 |
| Basal Leptin | 0.259 |
| Basal FFAs | 0.251 |
| Fasting insulin | 0.251 |
| ΔFFAs | 0.244 |
| Fasting leptin | 0.240 |
| ΔIGF1 | 0.229 |
| **Mouse** | |
| **Parameter** | **Average Pearson r** |
| Fasting HOMA-IR | 0.317 |
| Fasting insulin | 0.292 |
| ΔInsulin | 0.204 |
| ΔHOMA-IR | 0.203 |
| Basal BW | 0.203 |
| ΔBW | 0.202 |
| Basal HOMA-IR | 0.194 |
| Fasting β-HB | 0.192 |
| Basal insulin | 0.184 |
| ΔGlucose | 0.165 |

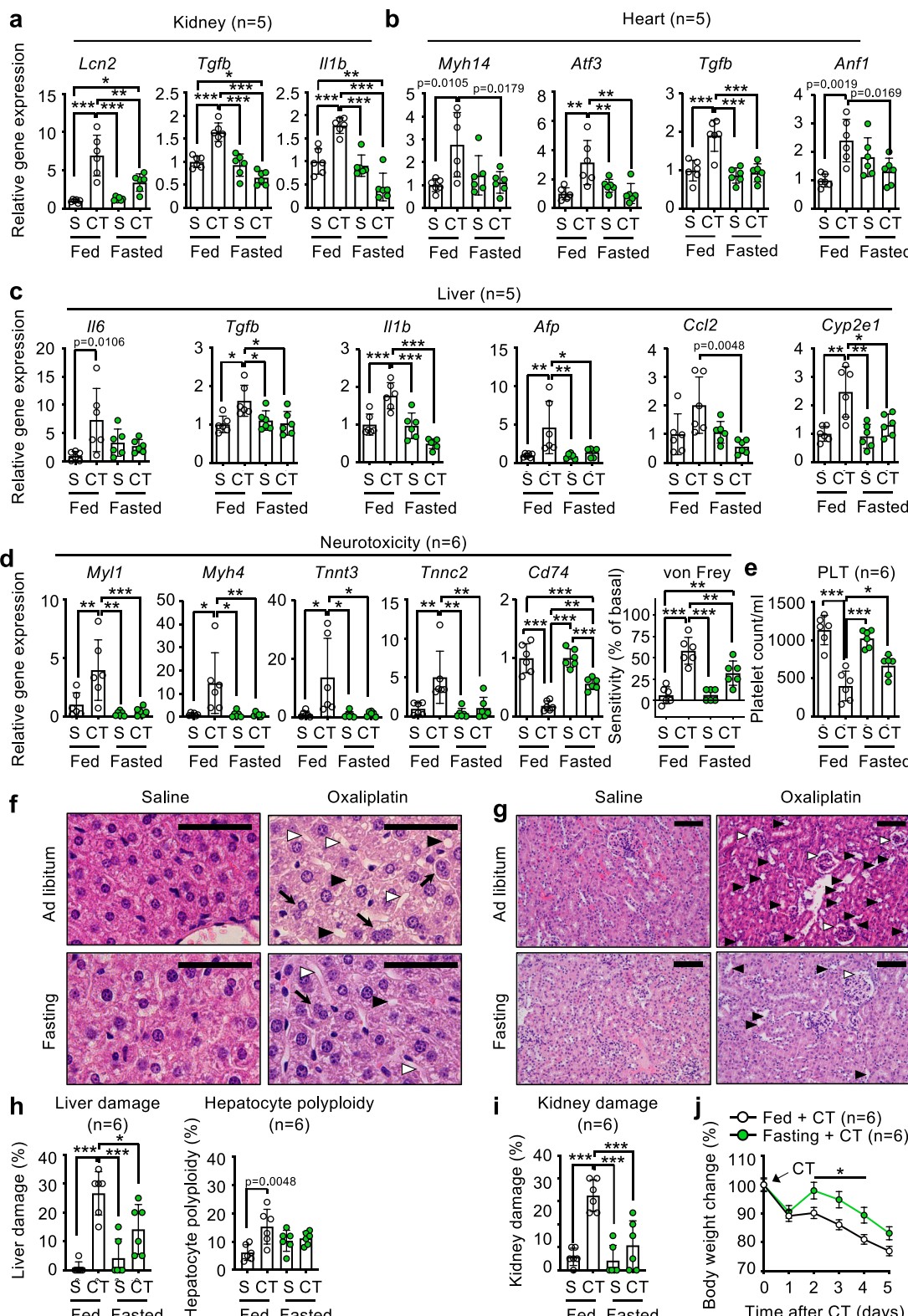

**Fig. 5 | Fasting protects from oxaliplatin-induced multi-organ toxicity in mice.** **a–e** Expression of the indicated toxicity-reporter genes in kidney (**a**), heart (**b**) liver (**c**), and dorsal root ganglia (**d**, panels to the left) or the neurotoxicity-measuring von Frey test (**d**, panel to the right); and blood platelets counts (**e**) from mice either fed ad libitum or fasted for 24 h before and 24 h after i.p. inoculation of saline (S) or 15 mg/kg oxaliplatin (CT), and sacrificed 5 days after chemotherapy administration. **f, g** Representative photographs of hematoxilin and eosin staining of sections from liver (**f**) or kidney (**g**) from the same mice as in (**a–e**), size bar = 200 μM. In liver (**f**), black triangles point to lipid vacuoles; white triangles point to sinusoidal dilatations; and black arrows point to polyploid hepatocytes. In kidney (**g**), black

triangles point to tubule dilatation, and white triangles point to Bowman capsule dilatation glomerulopathy. **h, i** Quantification of the histological findings from liver (**h**) or kidney (**i**). **j** Total body weight of mice from (**a–i**) at the indicated times after oxaliplatin (CT) administration. Bars and line-connected dots represent the average of the indicated number of mice per treatment group. Error bars represent the standard error of the mean. Statistical significance was assayed using the one-way ANOVA test (**a–e**) or the two-way ANOVA test (**j**) with Tukey correction for multiple comparisons. When possible, the exact $p$ value is provided for significant comparisons; otherwise, *$p < 0.05$; **$p < 0.01$; ***$p < 0.001$. Source data are provided as a Source Data file.

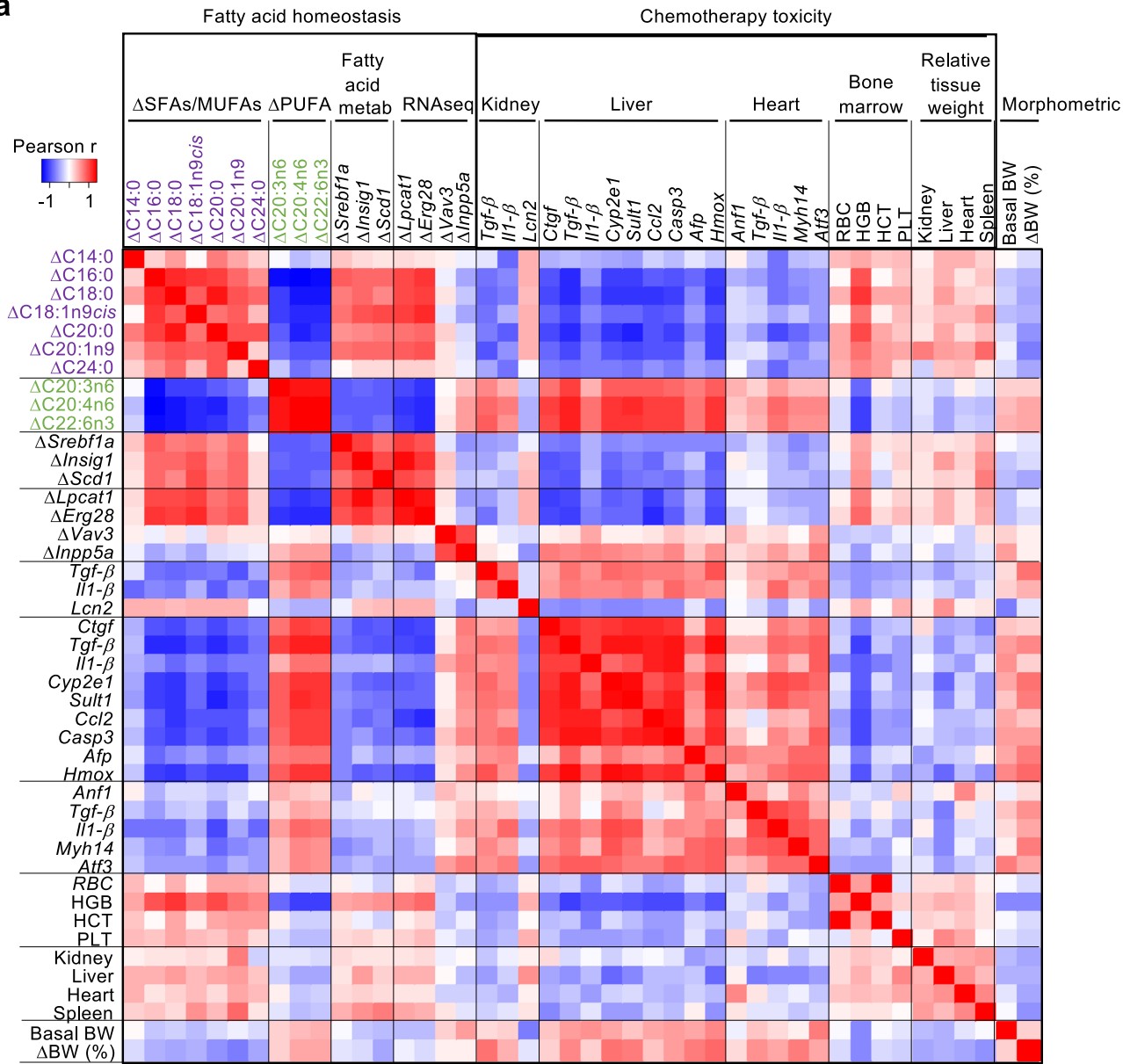

**Fig. 6 | Correlation of fatty acid homeostasis biomarkers with chemotherapy toxicity in mice. a** Heatmap depicting the Pearson correlation r coefficients between the indicated fatty acid homeostasis biomarkers (membrane fatty acids, *Srebf1* pathway genes in PBMCs and RNAseq-identified genes) with chemotherapy toxicity biomarkers: expression of toxicity-responding genes in kidney, liver and heart; bone marrow-associated parameters red blood cell counts (RBC), hemoglobin (HGB), hematocrit (HCT) and platelet counts (PLT); and tissue weights relative to total body weight. At the right end of the graph, the morphometric parameters of basal body weight (BW) and change in body weight with fasting in % (ΔBW (%)) are also analyzed. These correlations were performed with $n = 26$ mice. Source data are provided as a Source Data file.

good correlation between the fasting biomarkers and the relative tissue weight at sacrifice of liver, heart, kidney and spleen. We also studied the morphometrical parameters showing the strongest association with the fasting biomarkers described here: basal body weight and fasting-induced body weight loss, according to Table 3. Interestingly, we observed a marked direct correlation between these morphometric parameters and toxicity markers, although less strong than with the fasting biomarkers described here (Fig. 6a, right side of the diagram). Unfortunately, we had not measured insulinemia in all mice, so we could not include these biochemical parameters in this global analysis.

Next, we decided to generate a new mouse cohort to reproduce these results and test two crucial aspects of fasting and oxaliplatin toxicity: measure the response to fasting of other membrane lipid classes other than fatty acids; and associate these fasting biomarkers to neurological toxicity, one of the main toxicities elicited by oxaliplatin treatments ([57] and Fig. 5d). For this, we subjected a cohort of 21 male mice to a basal neurological von Frey test, followed by 24 h of fasting before blood sampling, a single intraperitoneal injection of 15 mg/kg oxaliplatin, followed by 24 h of fasting, then normal feeding for 3 days, a new von Frey neurological test, and finally sacrifice 5 days after chemotherapy to gather samples. For control, we used 9 littermate mice treated with oxaliplatin, but fed normally throughout the whole experiment. As already shown in Fig. 5, Fig. S6a, we first validated that fasting protected from chemotherapy toxicity in this experiment (Fig. S6b). We analyzed erythrocyte membrane fatty acids, as in previous figures, and determined the two groups that evolved most strongly in an opposite manner with fasting. We again observed,

as shown in Fig. 2, that one of these two groups was strongly enriched in SFAs/MUFAs (C18:0, C18:1, C20:0, C22:0, C24:0), and the other, opposite group was enriched in PUFAs (C22:6n3, C20:3n6, C20:5n3, C20:2n6). In addition, in the erythrocytes from the blood collected before and after the first 24 h of fasting and before oxaliplatin administration, we analyzed three families of membrane lipids: ceramides and hexosylceramides, representing lipids with significant signaling functions; and lysophosphatidylcholines, representing lipids of significant structural functions. Finally, 5 days after chemotherapy inoculation we measured many toxicity biomarkers of damage in kidney, liver, heart, bone marrow, nervous system, and morphometric parameters; and calculated the Pearson correlation coefficient between the changes in membrane lipids with fasting and the toxicity biomarkers. We confirmed once again that changes in membrane fatty acids strongly correlated with chemotherapy toxicity biomarkers in the same direction as in our previous experiment (Fig. S6d). In general, changes in ceramides (ΔCers) and hexosylceramides (ΔHexCers) with fasting did not correlate strongly with membrane fatty acids. ΔCers correlate directly with themselves, especially those with longer acyl chains, with the marked exception of ΔCer22:0. In turn, ΔHexCers correlate very strongly between themselves, with the marked exception of ΔHexCer24:0. Interestingly, ΔCers correlate inversely with ΔHexCers, suggesting opposite signaling functions of each lipid family during fasting. In the case of lysophosphatidylcholines (ΔLPC), the most evident finding was that PUFA-bearing ΔLPCs strongly correlate directly with themselves and with membrane ΔSFAs/MUFAs, and inversely with ΔPUFAs. With respect to chemotherapy toxicity biomarkers, ΔHexCers and ΔPUFA-LPCs were the lipids that most strongly correlated with toxicity, in both cases behaving in a similar way than ΔSFAs/MUFAs. However, ΔHexCers and ΔPUFA-LPCs correlate in different directions with toxicities in different organs: higher ΔHexCers and ΔPUFA-LPCs are associated with lower toxicity in kidney, liver, heart and bone marrow biomarkers; but their association with neurotoxicity is less clear. In general, neither ΔCers, ΔHexCers nor ΔLPCs correlated as clearly with chemotherapy toxicity as the membrane total fatty acids, that remain the best lipid biomarkers to predict chemotherapy toxicity.

### Lipid homeostasis biomarkers predict the fasting-mediated protection from chemotherapy toxicity

These findings suggest that, even though fasting protected all mice (Fig. 5, Fig. S6a, b), some fasting individuals were more protected than others. We had already defined two groups of individuals according to their fasting biomarkers (Fig. 4f). We wondered if these groups also presented different chemotherapy toxicity. For this, we determined for each individual mouse shown in Figs. 4, 6 if a given chemotherapy toxicity-related parameter was above or below the median of all the group. As shown in Fig. 7a, the same groups A and B that we had defined in Fig. 4f clearly defined two groups of response to chemotherapy: group A was more protected from oxaliplatin toxicity, and group B was less protected. Actually, there was a significant difference between both groups in most fasting and toxicity parameters: compared to group B, mice from group A showed a less strong decrease of SFAs and MUFAs (Fig. 7b, Fig. S7a) and a less strong increase in PUFAs (Fig. 7c, Fig. S7b); decreased less strongly the gene expression of the SREBP-1 pathway (Fig. 7d, Fig. S7c); showed a coordinated change with the insulin-responding genes (Fig. S7d); and showed a less-dramatic increase in heart (Fig. 7e, Fig. S7e), liver (Fig. 7f, Fig. S7f) and kidney (Fig. 7g, Fig. S7g) mRNA markers of toxicity. Mice from group A showed improved erythrocyte and platelet parameters (Fig. 7h, Fig. S7h). Fed mice treated with oxaliplatin showed a dramatic decrease in spleen size. Remarkably, mice from group A showed significantly larger spleen size after chemotherapy compared with mice from Group B. In addition, mice from group A showed a clear tendency for larger liver, kidney and heart (Fig. S7i). Histological analysis did not reveal any significant difference between groups A and B (Fig. S7j), indicating that

mRNA expression and hematological markers of toxicity were more sensitive than histological markers.

## Discussion

In this work we explored mechanisms driving the intense physiological and molecular response induced by short-term fasting in humans and mice. For this we used a study population of 20 healthy volunteers subjected to 36 h of fasting and 24 h of refeeding. In parallel, we studied a mouse cohort of 26 mice subjected to 24 h of fasting (Fig. 1a, b). We confirmed changes in classical blood biochemical parameters induced with fasting, such as decreased blood glucose and insulin; and increased FFAs and ketone bodies (Fig. 1c–i). In addition, we also observed decreased leptin and increased succinate in human blood, as already reported (Fig. S1d, e)[36]. Interestingly, accumulation of succinate has been found to induce thermogenesis and protect mice from high-fat diet induced obesity[83], an effect already observed in mice subjected to intermittent fasting[84]. The mechanisms driving increased succinate after short-term fasting are still unknown.

We have also studied the evolution of erythrocyte membrane fatty acids during fasting. The proportions of these fatty acids remained essentially unchanged with fasting in humans, with few exceptions. Moreover, correlations between changes in fatty acids during fasting in both mice and humans show a very interesting phenomenon: a coordinated change in membrane fatty acids, clearly defining one first group of PUFAs, that responded as a coordinated group, and a second group of SFAs and MUFAs, also evolving coordinately among them and in the opposite direction to the previous PUFAs group (Fig. 2c, d, Fig. S6c). Since the fatty acid levels are expressed as percentages of total fatty acids, it is normal that the increase in some fatty acids is accompanied by a decrease in others. The changes observed, however, were not random, but instead they followed a coordinated change of precise types of fatty acids. These results fit with previous reports showing that long-term fasting (7 days) in rheumatoid arthritis patients altered the membrane fatty acid composition of circulating platelets and neutrophils, although they did not distinguish between ΔPUFAs, ΔSFAs and ΔMUFAs and, therefore, they could not stratify their patients according to their fasting response[85]. Another interesting observation that we made is that changes in membrane fatty acids with fasting correlated with the basal status before fasting. This implies that individuals with lower levels of membrane PUFAs (and higher levels of SFAs and MUFAs) before fasting will likely change their membrane composition more dramatically. This correlation between basal fatty acids and their changes with fasting was more marked in mice (Fig. 2f) than in humans (Fig. 2e), which is probably a reflection of the much more homogeneous food composition in mice, and/or the different metabolic rates between both species. Interestingly, increased membrane PUFAs have been associated with decreased inflammation[39], at least partly by altering plasma membrane organization[86], and decreased membrane PUFAs have been associated to several aging-related diseases, as diabetes[87], Alzheimer's or Parkinson's diseases[88,89]. Therefore, our results indicate that short-term fasting changes the composition of cell membranes, enriching them in PUFAs, which is associated with better health status. This also applies to insulin-responding genes Srebf1, Scd1, Insig1, Lpcat1, and Erg28: they decrease with fasting. This evolution of insulin-responding genes is tightly associated with the evolution of membrane fatty acids: individuals with a higher basal expression of insulin-responding genes tend to have less basal PUFAs in their cell membranes, and both findings are associated to a worse metabolic status. When these individuals fast, they upregulate their PUFAs and downregulate their insulin-responding genes more strongly. We still do not know if these changes are permanent or reverse with time; or if repeated bouts of fasting have additive affects, a field of research that should be explored further.

Cell-free plasma circulating miRNAs are a very relevant source of information about physiological status in different settings such as

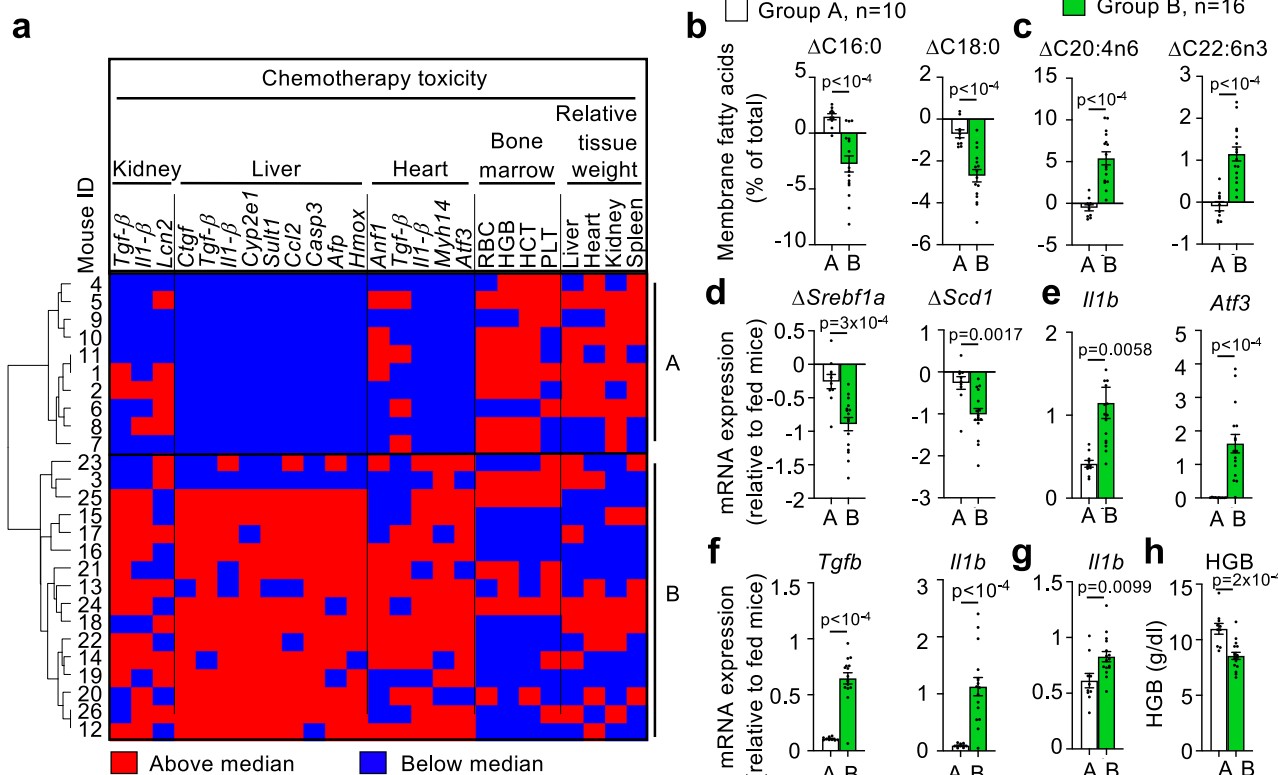

**Fig. 7 | Groups of individuals responding differently to fasting predict fasting-mediated protection from chemotherapy. a** Heatmap depicting the position of each individual (represented by a number next to each row) with respect to the median value for each of the indicated parameters. Red indicates "above the median", and blue indicates "below the median". Individuals are ordered according to the groups A and B found in Fig. 4f. RBC red blood cells. HGB hemoglobin. HCT hematocrit. PLT platelets. **b–h** Comparison between groups A ($n = 10$) and B ($n = 16$) of the indicated saturated (**c**) and polyunsaturated (**c**) erythrocyte membrane fatty acids; fatty acid metabolism genes (**d**); heart (**e**), liver (**f**) and kidney (**g**) toxicity-responding genes and bone marrow markers (**h**; hemoglobin, HGB). Bars represent the average of $n = 10$ (group A) and $n = 16$ (group B). Error bars represent the standard error of the mean. Statistical analysis was performed using the unpaired two-tailed Student $t$ test (**b–h**). The exact $p$ value is provided for each significant comparison. Source data are provided as a Source Data file.

cancer[90], cardiovascular disease[91] or general aging and age-related diseases[92]. One previous report on overnight fasting with 13 healthy humans showed no differences in miRNA profiles[41], but fasting times (12 h) were much shorter than those followed in the present study (36 h). In our study, we have found a set of 5 differentially expressed circulating miRNAs during our fasting protocol in humans. In all cases, we observed that changes elicited by fasting tended to be already apparent at 36 h, and to be maintained at least until the refed point. This indicated that the turnover of these miRNAs is longer than for other biochemical (Fig. 1) or genetic (Fig. 4) parameters. This discrepancy between our study and that by Max et al. suggests that time of fasting is a critical parameter when studying physiological and molecular changes, and reinforces the importance of moving away from general words like "fasting" and toward defining the exact period and type of fasting applied. In this regard, previous studies on the turnover of blood circulating tumor-associated miRNAs revealed a relatively long lifespan of around 14 days[93]. From these data, we hypothesize that some of the changes observed at the refeeding point could be reflecting changes already triggered during the fasting point. In our study, we focused our attention on miR-29b-3p and miR-100-5p that target lipid-related genes and constitute the first set of biomarkers for this fasting lipid-related response. Of interest, in mouse models, fasting-induced expression of miR-29b-3p[94] or its partner, miR-29c[95] in the liver, although the evolution of these miRNAs in the bloodstream during fasting was not reported. Interestingly, changes in human membrane fatty acids correlated very closely with the changes in levels of circulating miR-29b-3p and miR-100-5p that define almost exactly the same two groups of membrane fatty acids (Fig. 3c). In mice, we did

not observe the same evolution of miRNAs with fasting, again reflecting different regulation between humans and mice. We hypothesize that mice will most probably also suffer changes in their circulating miRNAs with fasting. Future research will reveal the exact identity of these altered miRNAs in mice, that should be correlated with our other fasting biomarkers to determine their functional relationship in the context of chemotherapy toxicity.

Also, when we analyzed which morphometric and biochemical parameters of our study showed the best correlation with ΔPUFAs, ΔSFAs and ΔMUFAs, and ΔmiRNAs, the best hits were HOMA-IR and insulinemia, both in mice and humans (Table 3). This goes in line with our findings with the pathway of the insulin-activated master regulator of fatty acid biosynthesis, SREBP-1 (Fig. 4a, b), that was consistently decreased in both humans and mice with fasting. Insulin signaling has previously been associated with membrane fatty acids. For example, patients with type-1 diabetes suffer a decrease in their membrane PUFAs and an increase in SFAs[87]. Also, fatty acid saturation has previously been shown to regulate SREBP-1 protein activation[96]. There are also previous reports linking the identified miR-29b-3p with insulin signaling and SREBP-1 activation: SREBP-1 induced the expression of the miR-29 family that, in turn, inhibited SREBP-1 expression[97]. Our work shows the tight relationship between changes during fasting in insulin signaling and changes in membrane fatty acids or circulating miRNAs in a comprehensive manner. These findings could help explaining the ability of intermittent fasting to improve insulin sensitivity[98]. We further corroborated this relationship by analyzing the mRNA expression profile of PBMCs between basal and fasting points. We identified a set of genes involved in the insulin signaling pathway, whose fasting-induced

changes strongly correlated with changes in membrane fatty acids in humans and mice (Fig. 4c, d), and with changes in miR-29b-3p and miR-100-5p in humans (Fig. 4c, panel to the right).

Our results comparing human and mouse fasting response reveal similarities (as blood glucose and ketone bodies, shown in Fig. S1a, b) but also discrepancies (most remarkably in body weight loss, as shown in Fig. S1c) that reflect a differential regulation in both species. Also, given their very different sizes and metabolic rates, mice and humans show different responses to the same times of fasting. Since the fasting response shows a sequential activation of molecular and physiological mechanisms, we hypothesize that we could also be detecting the fasting response in different phases between our mice and human samples. In any case, the discrepancies that we observe between both species would merit further research.

Our data also reveal several potentially important findings regarding the protective effects of fasting against chemotherapy toxicity. First, we confirm previous reports showing that fasting for 48 h protects from different chemotherapy treatments, such as oxaliplatin and doxorubicin (Fig. 5, Fig. S5a–g). Also, the minimum fasting time necessary for obtaining the best protection from chemotherapy toxicity is a clinically relevant issue. We explored the protective potential of shorter fasting times (24 h) around chemotherapy administration. Although we observed certain protection from several toxicity biomarkers with 24 h of fasting, the extent of this protection was not as clear as with 48 h of fasting (Fig. S5h–m, compared with Fig. 5). Many questions remain open in this regard: whether this minimum requirement of 48 h of fasting is also applicable to other chemotherapy treatments; if longer fasting times will protect even more; or if repeated fasting periods can improve protection. Further research in this regard will provide critical information for the translation of these results into the clinic.

In addition, using the fasting biomarkers described above, we could stratify participants in two clearly differentiated groups: groups A and B in Fig. 4e, f for humans and mice, respectively. We hypothesized from these data that the fasting lipid response that we describe here can predict the response to fasting of a given individual. Since fasting has been shown to affect the outcomes in different stress scenarios, such as aging-related markers[99], multiple sclerosis[100], inflammatory bowel disease[101], type 2 diabetes[9], ischemic stroke[102], or chemotherapy[22], we studied if our findings could help predict the response of patients to fasting in one of these scenarios: chemotherapy toxicity. Using our thorough phenotyping platform, we have been able to corroborate previous findings showing a strong protection from chemotherapy toxicity by 48 h of fasting in all tested tissues, as extensively shown in Fig. 5, Fig. S5. Importantly, we observed a very strong correlation between the fasting response and its ability to protect from chemotherapy toxicity in our mice: practically all toxicity biomarkers strongly correlated with the intensity of the fasting response, as shown in Figs. 6a, 7. Mice belonging to the group A, that showed a less-dramatic change in their membrane fatty acid composition and insulin-responding genes, were actually more strongly protected from chemotherapy toxicity. Also, these animals had lower basal body weight and lost less weight with fasting compared with group B. Of note, the kidney injury marker *Lcn2* was the only biomarker that behaved in an opposite direction compared with all the remaining toxicity biomarkers. Interestingly, this gene encodes for Lipocalin 2, that behaves as a driver of kidney disease progression[60]. Therefore, this opposite behavior of *Lcn2* in group A could reflect a slower kidney degradation in group A. These results provide an interesting insight: individuals with higher basal PUFAs and lower basal expression of insulin response genes, associated with better basal metabolic status before fasting, correlate with lower ΔPUFAs and higher ΔSFAs/ΔMUFAs (Fig. 4e, f), corresponding to Group A. Interestingly, this group presents the best protection from oxaliplatin toxicity, as shown in Figs. 6, 7. This indicates that individuals with the best basal metabolic status display a milder response to fasting in terms of membrane fatty acids, expression of insulin response genes, insulinemia and HOMA-IR; and they are the most protected from chemotherapy toxicity. Our findings have also relevant practical applications, because all these parameters could be improved with nutritional, behavioral or drug interventions before chemotherapy administration, improving the protecting effects of fasting in stressful scenarios such as chemotherapy administration.

We have extended the protective effects from oxaliplatin toxicity of 48 h of fasting in mice by treating mice with doxorubicin, an anthracycline unrelated to oxaliplatin (Fig. S5c–g). These results confirm a general protective effect of fasting against different chemotherapy stresses. Fasting had already been shown to protect from other chemotherapeutical treatments in mice, as etoposide[17,26], doxorubicin[24,103] or irinotecan[20]. Our complete toxicity panel confirms these previous results and allows for a thorough quantification of toxicity and correlation with fasting biomarkers in different scenarios.

Finally, our initial findings are based on correlations between different parameters during fasting and chemotherapy. However, these correlations have strong functional implications, since alterations in membrane composition and in the insulin signaling pathway are indeed exerting relevant functional changes, as those shown for membrane-bound PLA2 and SMase activites in liver in Fig. S2i, j. Even more importantly: since we can associate these correlations to chemotherapy toxicity, our findings have a functional application as biomarkers of fasting-mediated benefits, and can guide future applications of fasting in clinical and nutritional scenarios.

## Methods
### Animal experimentation
Animal experimentation at the National Center of Biotechnology (CNB, Madrid) was performed according to protocols approved by the CNB-CSIC Ethics Committee for Research and Animal Welfare (PROEX 148/18). Mice were housed between 18 and 24 °C and with 12 h dark/light cycle (light cycles from 7 a.m. to 7 p.m.), in constant humidity kept at 20–70%, and fed a standard chow diet, code TEK-LAD GLOBAL 2918.

For fasting and chemotherapy experiments, cohorts of 12–14-week-old male mice of the C57BL/6JOlaHsd background were generated. For mice shown in Figs. 1–4, 6, 7, 1 week before fasting, a basal sample of blood (200 μl) was taken in $K_2$EDTA tubes (Microvette CB 300 K2E, Sarstedt) from their submandibular vein at 9:00 a.m. to obtain basal PBMCs, erythrocytes and circulating miRNAs; and basal paw sensitivity was recorded using von Frey filaments, according to previous literature[76]. One week later, mice were fasted from 9:00 a.m. (2 h after lights on) for 24 h (Figs. 1–4, 6, 7) or 12 h (Fig. S5g–m) in clean cages with environmental enrichment, and blood was again taken in $K_2$EDTA tubes from their submandibular vein. Then, a single dose of oxaliplatin (15 mg/kg) was administered intraperitoneally. Mice were maintained under fasting conditions for 24 extra hours, amounting to a total fasting duration of 48 h. Four days after chemotherapy administration, a new measurement of paw sensitivity was recorded. Finally, 5 days after chemotherapy administrations, animals were sacrificed by $CO_2$, and tissues and blood samples were collected for analysis. A scheme for this animal protocol is shown in Fig. 1b.

For Fig. 5, the same protocol described above was performed, with the exception of the facial vein blood sampling before chemotherapy administration; and the administration of 15 mg/kg doxorubicin for Fig. S5c–g.

For hepatic SMase and cPLA2 activity determinations (Fig. S2), animal experiments were performed at the Spanish National Cancer Research Center (CNIO, Madrid), according to protocols approved by the CNIO-ISCIII Ethics Committee for Research and Animal Welfare (PROEX 162/14 and PROEX 315/16). Non-individualized C57BL/6JOlaHsd male mice of 10–15 weeks of age were fasted for 24 h (from 9:00 a.m., 2 h after lights on) in clean cages with environmental enrichment, then were sacrificed, and samples collected for analysis.

## Human nutritional trial

**Ethics and inclusion and exclusion criteria.** The human study protocol was approved by the Research Ethics Committee of the Madrid Institute for Advanced Studies (IMDEA) Food Foundation (PI-0025), according to the standards of the Helsinki Declaration, and registered in the ClinicalTrials public portal, ClinicalTrials.gov Identifier: NCT04259879. This protocol meets the definition of Basic Experimental Studies Involving Humans (BESH) since it is a systematic study directed toward greater understanding on how fasting changes several human body parameters with no intention to change the health status of the participants. Data and samples were collected at the GENYAL platform, in the IMDEA Food institute headquarters, in June 2016. Primary outcomes included gene expression measurements in PBMCs: expression analysis of different genes from PBMCs were performed in a HT-7900 Fast Real-time polymerase chain reaction (PCR) System (Applied Biosystems). Quantifications were made applying the $\Delta Ct$ method ($\Delta Ct = [Ct$ of gene of interest $- Ct$ of housekeeping]). The housekeeping genes used for input normalization were $\beta$-actin (*ACTB*) and ribosomal protein lateral stalk subunit P0 (*RPLP0*). Secondary outcomes included blood hormones and metabolite measurements (insulin, free fatty acids, ketone bodies, leptin, lipid profiles), using standardized protocols. Also, for a subjective evaluation of tolerance to fasting, participants filled in a fasting tolerance test based on the symptoms they feel, that resulted in a final score of tolerance to fasting. A group of 20 volunteers, 10 men and 10 women, were recruited via the Cantoblanco Platform for Nutritional Genomics and Food "GENYAL", promoted by IMDEA Food. All participants signed an informed consent form before enlisting in the trial, accepting publication of results. An economic compensation for their participation was given to each participant. Sample size was estimated from previous reports[6,32–34]. Individuals who were taking weight loss drugs, supplements or psychoactive substances were excluded, as were those who had any condition that might interfere with short-term fasting, e.g., severe psychiatric disease, uncontrolled hyperthyroidism, diabetes mellitus, associated food disorders, treated dyslipidemia or hypertension, with BMI <19 kg/m$^2$ or >30 kg/m$^2$; with abnormal fasting glucose levels; who had donated blood less than 8 weeks before the study; and being either pregnant or breastfeeding. The personal clinical history of each volunteer was checked for any other disease, but unless an active disease was confirmed (e.g., hepatic, renal or cardiovascular disease that requires special treatment and ongoing monitoring) it was not considered a reason for exclusion.

**Fasting and refeeding intervention.** For the fasting and refeeding experiment, volunteers were instructed to follow their normal feeding pattern until dinner at 9:00 p.m. of day 0, and then began their fasting period in their own homes, during which plain water was allowed ad libitum, but no solid food or beverages other than water were allowed. After 12 h of fasting, blood samples were taken at IMDEA Food at 9:00 a.m. on day 1 (overnight fasting, "Basal" point), and patients returned to their usual activities until 9:00 a.m. on day 2, when a second blood sample was taken at IMDEA Food (36 h fasting, "Fasting" point). Volunteers then ate breakfast and continued a normal feeding pattern and activity during day 2 including normal dinner 9:00 p.m., and then they began an overnight fasting period until 9:00 a.m. on day 3. At this point another blood sample was taken at IMDEA Food (overnight fasting, "Refed" point) and volunteers resumed their normal feeding patterns and activity. Fasting and refeeding times are similar to previous reports in humans[6,32–34], allowing for comparison of results.

## Obtention of anthropometric, dietetic and blood cardiovascular data

**Anthropometric variables.** Anthropometric data were obtained as follows: weight (kg), body-mass index (BMI, kg/m$^2$), total fat mass (TFM%) and total muscular mass (TMM%) and visceral fat were measured in light clothing and without shoes to the nearest 0.1 kg using a BF511 Body Composition Monitor (Omron Healthcare Co. Ltd., Kyoto, Japan). Height was measured to the nearest half centimeter using a wall-mounted Leicester stadiometer (Biological Medical Technology SL, Barcelona, Spain). BMI was calculated as weight in kilograms divided by height in meters squared. Waist circumference was measured to the nearest 0.5 cm using an anthropometric tape midway between the lowest rib and at the iliac crest at minimal respiration using a Seca 201 non-elastic tape (Quirumed, Valencia, Spain).

**Blood pressure.** Systolic (SBP) and diastolic blood pressure (DBP) were measured using a Model M3 Automatic Digital Blood Pressure Monitor (Omron Healthcare Co. Ltd., Kyoto, Japan). Measurements were made at baseline of the study with the volunteers comfortably seated and having neither eaten nor exercised in the previous 30 min. A minimum of three readings were taken at intervals of at least 1 min, and the mean SBP and DBP calculated.

**Sampling and storage.** Blood samples were taken early in the morning at IMDEA Food in 9 ml or 2.6 ml EDTA tubes (S-Monovette® 9 ml K3E, Sarsted 02.1066.001 or S-Monovette® 2.6 ml K3E, Sarsted 04.1901.100) at the different time points indicated above. Nine ml of the blood samples were kept at room temperature for PBMCs purification, and the rest was kept at 4–6 °C for biochemical analysis, which was always performed within 48 h of sample extraction. For plasma and erythrocyte isolation, 2.6 ml of the blood samples were centrifuged at $1000 \times g$ for 10 min and the packed erythrocytes and the plasma were immediately stored at −80 °C until analysis.

## Laboratory measurements

**Human and mouse blood analysis.** Human blood was analyzed by Laboratory CQS Consulting for glucose (kit reference 3L8221) and FFAs (kit references 434-91795 and 436-91995), all from Abbott Laboratories, by enzymatic spectrophotometric assays using an Architect CI8200 instrument (Abbott Laboratories, IL, USA). Insulin (kit reference 8K4127, Abbott Laboratories) was determined by luminescent immunoassay using the same Architect CI8200 device, and insulin resistance determined using the HOMA-IR = [Fasting glucose (mg/dl) x Fasting insulin ($\mu$U/ml)]/405. Leptin (kit reference 10-1199-01, Mercodia) was measured by non-competitive automatic ELISA immunoanalysis. $\beta$-hydroxybutyrate (kit reference MAK041, Sigma-Aldrich) was measured by an enzymatic spectrophotometric assay using a Multiskan equipment (Thermo Fisher). Succinate was measured using the EnzyChromTM Succinate Assay Kit (BioAssay Systems, ESNT-100).

Mouse blood was collected from tail to measure glucose, ketone bodies and insulin. Glycemia and ketonemia were determined using the Nova Pro GLU/KET system from Menarini Diagnostics (Statstrip Glucose Test Strips, 40493 and Statstrip B-ketones Test Strips, 41543). Insulinemia was determined using the Ultra Sensitive Mouse Insulin ELISA Kit (Crystal Chem. Cat No. 90080). For hemograms, blood samples were collected in EDTA tubes by cardiac puncture after $CO_2$-sacrifice of the mouse and analyzed using a Laser Cell blood counter (CVM veterinary diagnostic).

**PBMCs isolation from mouse and human blood.** Blood samples used for PBMCs isolation were processed in the first 1–2 h after collection. PBMCs from mouse and human whole blood were isolated using Histopaque®-1077 (Sigma-Aldrich, 10771). Briefly, 9 ml of blood from human volunteers or 100–200 µl of blood from mice were diluted 1:2 with PBS and layered onto Histopaque (half volume for human samples, and 400 µl for mouse samples) at room temperature. After centrifugation at $500 \times g$ for humans, or $400 \times g$ for mice during 30 min at room temperature at the lowest acceleration and brake, the opaque interface containing the PBMCs was transferred with a Pasteur pipette

to a clean tube, washed three times with PBS and the remaining pellet was resuspended with Tri-Reagent and kept at −80 °C until all samples were processed.

**qRTPCR.** Total RNA was extracted using Tri Reagent (Molecular Research Center, MRC-TR-118), and, for PBMCs, it was further processed using an RNA purification kit (DirectZol RNA microprep, Zymo Research R2060). Samples were reverse transcribed using random priming and High-Capacity cDNA Reverse Transcription kit (Applied Biosystems™, 4368814), according to the manufacturer's instructions. Quantitative real-time PCR was performed using GoTaq qPCR Master Mix (Promega, A6002) in an ABI PRISM 7900 thermocycler (Applied Biosystems). Quantifications were made applying the ΔCt method (ΔCt = [Ct of gene of interest − Ct of housekeeping]). The housekeeping genes used for input normalization were *ACTB (β-actin)* and *RPLP0* (also known as *36b4* in mice). Primer sequences are the following:

| | |
|---|---|
| Mm-*β-actin* | Fwd: GGACCACACCTTCTACAATG |
| | Rvs: GTGGTGGTGAAGCTGTAGCC |
| Mm-*36b4* | Fwd: AGATTCGGGATATGCTGTTGG |
| | Rvs: AAAGCCTGGAAGAAGGAGGTC |
| Mm-*Cpt1a* | Fwd: CAGCTCGCACATTACAAGGA |
| | Rvs: GAAGAGCCGAGTCATGGAAG |
| Mm-*Slc25a20* | Fwd: TGCCAGTGGGATGTATTTCA |
| | Rvs: GTTGAAGATCCCTGCAAAGC |
| Mm-*Plin2* | Fwd: ATTCTGAACCAGCCAACGTC |
| | Rvs: CTTATCCACCACCCCTGAGA |
| Mm-*Srebf1a* | Fwd: CGGGGAACTTTTCCTTAACGTG |
| | Rvs: CTCTCAGGAGAGTTGGCACC |
| Mm-*Scd1* | Fwd: TCCCGGGAGAATATCCTGGTT |
| | Rvs: ACAGGAACTCAGAAGCCCAAAG |
| Mm-*Insig1* | Fwd: CCTTGACTTTAGCAGCCCTCT |
| | Rvs: TCGTCCTATGTTTCCCACTGT |
| Mm-*Inpp5a* | Fwd: TGCAGAAGAACTGGCTTCGG |
| | Rvs: GACATGGAGGCCTCGTAGTTT |
| Mm-*Vav3* | Fwd: GAGCTTCCAGACAGGTGGAC |
| | Rvs: GCCTGCCTGGATATGCAATG |
| Mm-*Erg28* | Fwd: CTCTACACTGGCAAGCCAAAC |
| | Rvs: TGAGAGCAGCGTCCAGATCC |
| Mm-*Lpcat1* | Fwd: CAGGACTCGCGAAGGAAGAC |
| | Rvs: ACTCCAGGAATGAACGCACC |
| Mm-*Atf3* | Fwd: TACCGTCAACAACAGACCCC |
| | Rvs: CCGCCTCCTTTTCCTCTCAT |
| Mm-*Myh14* | Fwd: TGATCGCTGCTCTGGAGTCT |
| | Rvs: GAACTACCTCCTTCAGCCGC |
| Mm-*Lcn2* | Fwd: TGTCACCTCCATCCTGGTCA |
| | Rvs: ACTGGTTGTAGTCCGTGGTG |
| Mm-*Hmox1* | Fwd: AGCCCCACCAAGTTCAAACA |
| | Rvs: AAGTGACGCCATCTGTGAGG |
| Mm-*Afp* | Fwd: GCTGCTCAGTACGACAAGGT |
| | Rvs: TGGTTGTTGCCTGGAGGTTT |
| Mm-*Casp3* | Fwd: GAGCTGGACTGTGGCATTGA |
| | Rvs: GCTGCAAAGGGACTGGATGA |
| Mm-*Ccl2* | Fwd: CCACTCACCTGCTGCTACTC |
| | Rvs: GGACCCATTCCTTCTTGGGG |
| Mm-*Sult1c2* | Fwd: GCCAGACGACCTCCTCATTT |
| | Rvs: GCTGAGTGGGAAGATGGGTC |

| | |
|---|---|
| Mm-*Cyp2e1* | Fwd: GACCACCAGCACAACTCTGA |
| | Rvs: CTCATGCACTACAGCGTCCA |
| Mm-*Anf1* | Fwd: ATCTGCCCTCTTGAAAAGCA |
| | Rvs: ACACACCACAAGGGCTTAGG |
| Mm-*Sox2* | Fwd: ACGCCTTCATGGTATGGTCC |
| | Rvs: GCTTCTCGGTCTCGGACAAA |
| Mm-*Il6* | Fwd: GTTCTCTGGGAAATCGTGGA |
| | Rvs: GGTACTCCAGAAGACCAGAGGA |
| Mm-*Il1b* | Fwd: AAAAGCCTCGTGCTGTCG' |
| | Rvs: AGGCCACAGGTATTTTGTCG |
| Mm-*Tgfb* | Fwd: TGCGCTTGCAGAGATTAAAA |
| | Rvs: CTGCCGTACAACTCCAGTGA |
| Mm-*Ctgf* | Fwd: AGAACTGTGTACGGAGCGTG |
| | Rvs: TGCTTTGGAAGGACTCACCG |
| Mm-*Myl1* | Fwd: TTGGGAACCCCAGCAATGAA |
| | Rvs: ACGCAGACCCTCAACGAAAT |
| Mm-*Myh4* | Fwd: CTGGGGGACAAGCTGCG |
| | Rvs: ATCTGACAAAGTGGGGGTGG |
| Mm-*Tnnc2* | Fwd: GTGCTTCCGCATCTTTGACA |
| | Rvs: TGAACGCCCTCCATCATCTT |
| Mm-*Tnnt3* | Fwd: ATTGACCAAGCCCAGAAGCA |
| | Rvs: GAAGGCTGGAGAGTGCAGAG |
| Mm-*Cd74* | Fwd: TGGATGGCGTGAACTGGAAG |
| | Rvs: TCTTCCTGGCACTTGGTCAG |
| Hs-*β-ACTB* | Fwd: CAAGGCCAACCGCGAGAAGAT |
| | Rvs: CCAGAGGCGTACAGGGATAGCAC |
| Hs-*RPLP0* | Fwd: TCATCAACGGGTACAAACGA |
| | Rvs: GCCTTGACCTTTTCAGCAAG |
| Hs-*CPT1a* | Fwd: CCCATTCGTAGCCTTTGGTA |
| | Rvs: AAAACTTGCCCATGTCCTTG |
| Hs-*SLC25A20* | Fwd: TGGGACCTTTGACTGTTTCC |
| | Rvs: CCGTGATGCCCTCTCTAAAA |
| Hs-*PLIN2* | Fwd: AAGAAAAATGGCATCCGTTG |
| | Rvs: ATACGTGGAGCTCACCAAGG |
| Hs-*SCD* | Fwd: TTCCCGACGTGGCTTTTTCT |
| | Rvs: AGCCAGGTTTGTAGTACCTCC |
| Hs-*SREBF1* | Fwd: AGCCATGGATTGCACTTTCG |
| | Rvs: CCAGCATAGGGTGGGTCAAA |
| Hs-*INSIG1* | Fwd: CACGGTGGGGAACATAGGAC |
| | Rvs: TGCTCTCAGAATCGGTGGTT |

**RNAseq.** Library construction and sequencing were performed by the Novogene Bioinformatics Institute (Beijing, China). After mRNA enrichment from organisms using oligo(dT) beads, library preparation was performed with NEB Next Ultra RNA Library Prep Kit. First, the mRNA is fragmented randomly by adding fragmentation buffer, then the cDNA is synthesized by using mRNA template and random hexamers primers, after which a custom second-strand synthesis buffer (Illumina), dNTPs, RNase H and DNA polymerase I are added to initiate the second-strand synthesis. Second, after a series of terminal repair, a ligation and sequencing adaptor ligation, the double-stranded cDNA library is completed through size selection and PCR enrichment. The qualified libraries are fed into Illumina sequencers (Novaseq PE150) after pooling according to its effective concentration and expected data volume.

For data analysis, HISAT2 v.2.1.0[104] was used to align paired end reads against the human "genome_snp_tran" index version GRCh38p12, which includes sequences from the reference genome plus SNPs and transcripts. The RNA strandness parameter was set to

"FR". Rates of paired reads aligned concordantly exactly 1 time was higher than 86% for all samples. StringTie v.1.3.5 was used to assemble aligned reads into potential transcripts, and those were annotated based on GRCh38 v.94 genomic features. Prior to differential expression analysis, read count information from all samples was extracted using a custom Python v3.7.0 script.

For differential gene expression analysis, an additive linear model, containing sample pairing information in addition to the treatment factor, was used within EdgeR Bioconductor software package v3.21.9. Reactome database was used for pathway functional analysis in conjunction with Gene Ontology gene annotations.

**Isolation and analysis of circulating miRNAs.** Circulating miRNA isolation and analysis were performed following the protocols and advice from Exiqon and Bionova. Total RNA, including miRNAs, was isolated from 200 µl of plasma of human volunteers or from 60 µl of plasma of mice, using the miRNeasy Serum/Plasma Advanced Kit (Qiagen, Cat No.: 217204). For reverse transcription of the isolated miRNAs, the miRCURY LNA RT Kit (Qiagen, Cat.no. 339340) was used. The efficiency of RNA isolation, synthesis of cDNA and qPCRs was monitored using the RNA Spike-in kit, UniRT (Qiagen, Cat No.: 339390), including the miRNAs UniSp2, UniSp4 and UniSp5.

For the exploratory screening, cDNAs from 4 volunteers at two time points (basal and fasted conditions) were analyzed using the Human Serum/Plasma miRCURY LNA miRNA Focus PCR Panel (Qiagen, Cat. no. YAHS-106YE-2: 384-well format), containing 192 miRNA-specific Locked Nucleic Acid (LNA)-enhanced primers. The qPCR reaction was performed using the master mix miRCURY LNA SYBR Green PCR Kit (Cat No.: 339347) in an ABI PRISM 7900HT equipment (Applied Biosystems). Each 384-well plate of the miRNA Focus PCR Panel was used for two cDNA samples of the same volunteer at the two different time points, basal and fasting states. No replicates were used for this first screen.

For validation of all available human samples circulating miRNAs and analysis of mouse circulating miRNAs, 2 µl of the total RNA were reverse transcribed, and the resulting cDNA was analyzed, as described previously for the exploratory screening, using the following LNA PCR primer sets:

hsa-miR-191-5p (Qiagen, Cat No.: YP00204306).
hsa-miR-425-3p (Qiagen, Cat No.: YP00204038).
hsa-miR-103a-3p (Qiagen, Cat No.: YP00204063).
hsa-miR-93-5p (Qiagen, Cat No.: YP00204715).
hsa-miR-223-3p (Qiagen, Cat No.: YP00205986).
hsa-miR-30c-5p (Qiagen, Cat No.: YP00204783).
hsa-miR-16-5p (Qiagen, Cat No.: YP00205702).
hsa-miR-425-5p (Qiagen, Cat No.: YP00204337).
hsa-miR-423-5p (Qiagen, Cat No.: YP00205624).
hsa-miR-100-5p (Qiagen, Cat No.: YP00205689).
hsa-miR-29b-3p (Qiagen, Cat No.: YP00204679).
hsa-miR-194-5p (Qiagen, Cat No.: YP00204080).
hsa-miR-485-3p (Qiagen, Cat No.: YP00206055).
hsa-miR-141-3p (Qiagen, Cat No.: YP00204504).
UniSp2 (Qiagen, Cat No.: YP00203950).
UniSp4 (Qiagen, Cat No.: YP00203953).
UniSp5 (Qiagen, Cat No.: YP00203955).
mmu-miR-16-5p (Qiagen, Cat No.: YP00205702).
mmu-miR-30e (Qiagen, Cat No.: YP00204714).
mmu-miR-146a-5p (Qiagen, Cat No.: YP00204688).
mmu-miR-195 (Qiagen, Cat No.: YP00205869).

For analysis of the miRNA qPCR reactions, sample data points with a Ct > 36 were excluded from data analysis. ΔCt and ΔΔCt were calculated using the following mathematical formula: ΔCt = Ct Endogenous reference (average of all reference circulating miRNAs) − Ct sample, and ΔΔCt = ΔCt fasting − ΔCt basal. We performed ΔCt normalization using the most stable plasma miRNAs as reported in the literature[105–107] and also following manufacturer's recommendations. The final list of endogenous reference circulating miRNAs for humans was: hsa-miR-16-5p, hsa-miR-30c-5p, hsa-miR-93-5p, hsa-miR-103a-3p,

hsa-miR-191-5p, hsa-miR-223-3p, hsa-miR-423-5p, hsa-miR-425-3p and hsa-miR-425-5p. For mice, endogenous reference circulating miRNAs were mmu-miR-16-5p, mmu-miR-30e-5p, mmu-miR-146a-5p, and mmu-miR-195a-5p.

Human circulating miRNAs with significant differences between the collection points were selected for further functional analyses. Enrichment analysis was carried out with DIANA miRPath v0.3[108] using TarBase v7.0 database and a False Discovery Rate adjustment. Over-represented KEGG pathways are shown in heatmaps with the color representing the log10 of the $p$ value. Experimentally validated and predicted targets of these miRNAs were searched using the miRWalk tool[109]. Predicted targets were analyzed with miRWalk comparing 9 algorithms (miRanda, TargetScan, miRDB, PITA, MicroT4, miRMap, microRNAMap, miRWalk and RNA22) using stringent criteria in all of them. Predicted targets by, at least, 7 algorithms were selected for further analysis.

Mouse circulating miRNAs were analyzed following the same procedures as described above for human circulating miRNAs, using the adequate mouse endogenous references.

**Fatty acid measurement from erythrocytes.** The erythrocyte membrane fatty acid profile was determined by gas-chromatography as described[110]. In brief, cells contained in a 100 µl of packed erythrocytes were hemolyzed and centrifuged. The pellet was dissolved in 1 ml $BF_3$ methanol solution and heated to hydrolyze and methylate glycerophospholipid fatty acids. The fatty acid methyl esters were isolated by adding n-hexane. For tissues (brain, heart, liver, and kidney), the fatty acid profiling of phospholipid fraction was performed as previously described[111]. Briefly, frozen tissue samples (~10 mg) were powdered under liquid nitrogen. Total lipids were extracted with chloroform and methanol and were evaporated under a stream of nitrogen (37 °C). Phospholipids were sequentially isolated by solid-phase extraction using 100 mg Bond Elut aminopropyl silica cartridges (Varian Sample Preparation Products). Acylated fatty acids were converted into their respective methyl esters by incubation with methanol containing 2% (v/v) $H_2SO_4$. The extracts were cooled and fatty acid methyl esters were isolated by adding n-hexane. In both cases, fatty acid methyl esters were separated by gas-chromatography using an Agilent HP 7890 Gas Chromatograph equipped with a 30 m × 0.25 µm × 0.25 mm SupraWAX-280 capillary column (Teknokroma, Barcelona, Spain), an autosampler, and a flame ionization detector. In all cases, the amount of each fatty acid was expressed as a percentage of the total identified fatty acids in the sample, as recommended in the literature[112].

**Lipidomic profile analysis.** Concentrations of lipids (ceramides, hexosylceramides and lysophosphatidylcholines-LPCs) were determined by liquid chromatography coupled to tandem mass spectrometry (LC−MS/MS), as previously described[113] with slight modifications. Briefly, 50 µL of sample were spiked with 200 µL of methanolic ice-cold and 50 µl of internal standard solution. After vortexing and centrifugation (5 min, 2917 × $g$, 4 °C), the supernatant was transferred to a HPLC vial and 5 µl were injected into LC−MS/MS system.

The chromatographic separation of the lipid species was performed using an Acquity UPLC instrument (Waters Associates, Milford, Massachusetts, USA) equipped with an Acquity UPLC® (BEH C18, 1.7 µm, 2.1 × 100 mm) column (Waters Associates). The flow rate was 0.3 ml/min and the temperature of the column was set at 55 °C. An isocratic method was used, with a solution of 1 mM ammonium formate and 0.01% formic acid in methanol as the mobile phase solvent. The detection of the lipid species was performed with a triple quadrupole (Xevo TQS-Micro MS, Waters) mass spectrometer equipped with an electrospray ionization source operated in the positive ion mode. The monitoring and quantification of the lipids was performed in the multiple reaction monitoring mode.

**Cytosolic phospholipase A$_2$ activity**. Cytosolic phospholipase A$_2$ activity (cPLA2 activity) was measured in livers from control (fed ad libitum, $n = 6$) and fasting mice (24 h fasting, $n = 6$) using the Cytosolic phospholipase A$_2$ assay kit from Abcam (ab133090) according to manufacturer instructions. Briefly, 15 to 30 mg of mouse liver tissue were homogenized in Hepes/EDTA cold buffer (50 mM Hepes, pH 7.4, 1mM EDTA) with a homogenizer T10 basic ULTRA-TURRAX-IKA (100 μl buffer *per* 10mg of tissue). After centrifugation at 10,000 × *g* for 15 min at 4 °C, 100 μl of each supernatant were dialyzed using Thermo Scientific™ Slide-A-Lyzer™ 20K MWCO MINI Dialysis Devices (Thermo Scientific™ 88402), first for 2 h at RT and after changing the dialysis buffer, overnight at 4 °C. All samples were incubated before assaying with 5 μM Bromoenol lactone for 15 min at 25 °C to inhibit Ca$^{2+}$-independent phospholipase (iPLA2) activity. Triplicates of each sample (10 μl) were used to determine cPLA2 activity after 60 min of reaction. Final absorbance at 405 nm was read using a VICTOR Nivo Multimode Microplate Reader (Perkin Elmer) and the cPLA2 activity was calculated according to manufacturer's instructions.

**Neutral sphingomyelinase activity**. Neutral sphingomyelinase activity (nSMase activity) was measured in livers from control (fed ad libitum) and fasting mice (24 h fasting) using the colorimetric Sphingomyelinase Assay kit from Abcam (ab138876) according to manufacturer instructions. Briefly, 10 to 20 mg of liver tissue were homogenized using a tissue homogenizer (T10 basic ULTRA-TURRAX-IKA) in the mammalian cell lysis buffer, ab179835, containing protease and phosphatase inhibitors (1mM phenyl-methyl-sulfonyl fluoride -PMSF-, protease inhibitor cocktail from Sigma-Aldrich P8340, and 1mM NaF and 10mM sodium orthovanadate, respectively). 200 μl of lysis buffer were added *per* 10 mg of tissue. After centrifugation at 10,000 × *g* for 10 min at 4 °C, protein concentration was determined using *DC*™ protein assay, (BioRad 5000111) and lysates were diluted to 6.4 mg/ml with lysis buffer. After testing several dilutions of the protein extracts to quantify their sphingomyelinase activity, a 2:3 dilution with assay buffer was found optimal for the quantification of the SMase activity by measuring A655 after 1 h of reaction using a VICTOR Nivo Multimode Microplate Reader (Perkin Elmer). 50 μl of each sample were measured in duplicates.

**Mouse histopathology**. Liver, heart and kidney were dissected from mice, fixed in formalin and embedded in paraffin blocks. 5 μm-thick sections were stained with hematoxylin and eosin.

Tissue histological characteristics were assessed following Nonneoplasic Lesion Atlas, performed by the National Toxicology Program (https://ntp.niehs.nih.gov/nnl/index.htm). Two representative fields of ×20 magnification were analyzed for each slide. Median liver lobes were analyzed for the occurrence of hepatocyte polyploidy, incidence of lipid vacuoles and sinusoidal dilatation. Longitudinal planes of kidney were analyzed for the occurrence of dilatation of Bowman space and tubular dilatation. Transversal sections of heart were analyzed for cardiomyopathies as multifocal necrosis, small number of cardiomyocytes and myofiber vacuolation; hemorrhages and infiltration. For hepatocyte polyploidy, both total and polyploid nuclei were counted, and the polyploid/total ratio was calculated. For the rest of the histological findings, a subjective quantification of the level of damage was assigned to each slide, taking into account the histological findings indicated above. All microscopic evaluations (10–40X magnification) were performed in a blinded manner using the DM2000 LED optical microscope with the DMC5400 Research camera (Leica).

**Statistical analyses**
Statistical analyses were performed using GraphPad Prism version 8.4.3 for Mac OS X and R version 4.0.2 (https://www.r-project.org/). Missing data (2 items from *ITGB8* expression; 2 items from the hsa-miR-29b-3p; and 2 items from the hsa-miR-100-5p) were singly imputed by means of the missForest R package. Mean differences between two groups were analyzed using Student's two-tailed *t* test. Differences between three or more groups were analyzed using one-way ANOVA followed by post hoc analysis through Tukey's multiple comparison test. Both analyses were performed as paired (repeated measures in the case of ANOVA) or unpaired tests, depending on the type of analyzed samples. Linear correlations were performed using the Pearson test. Data distribution was considered normal if it passed at least two out of three normality checks (D'Agostino & Pearson omnibus, Shapiro-Wilk and Kolmogorov-Smirnov normality tests).

For the systematic analysis of correlations between variables, full Pearson correlation matrices were generated. The difference between Fasting and Basal reads of all variables (biochemical, fatty acids, gene and miRNA expression) were used, together with morphometric variables. The matrices were plotted as correlation plots or heatmaps, with variables clustered based on a hierarchical Ward method[114,115].

For the analysis of trends of changes in variables vs individuals (Figs. 4e, f, 7a), variables were assigned a 1 value if the change was above the median, or −1 if they were below or equal to the median. An Euclidean distance was then calculated between all the individuals using these variables, and they were clustered by a Ward's hierarchical method, and cutting the dendrogram at the two clusters level.

In order to identify the morphometric and biochemical variables that best correlated with the described fasting biomarkers (membrane fatty acids, circulating miRNA and PBMC gene expression), we calculated the average absolute value of the Pearson correlation of each morphometric and biochemical variable with the fasting biomarkers, and then ranked in decreasing order, as presented in Table 3.

**Reporting summary**
Further information on research design is available in the Nature Research Reporting Summary linked to this article.

## Data availability
The datasets generated in this study have been deposited in the Zenodo database under the accession code https://doi.org/10.5281/zenodo.6958418; and in the GEO database under the accession code GSE173241. Source data for each figure panel are provided with this paper.

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

## Acknowledgements

General: We thank Prof. José María. Ordovas for his kind suggestions; nutritionists Helena Marcos-Pasero, Elena Aguilar-Aguilar and Isabel Espinosa-Salinas for their help with volunteers management; Rosa Serrano for her help with animal experiments; Susana Molina for her advice with PBMC isolation; Luisa Mariscal, Domingo Fernández, Lola Martínez, Diego Megías, Patricia Gonzalez, Fernando Peláez, Anabel Sanz, Carolina Pola, Celia de la Calle, Ana Ortega, Ana Sagrera, José Miguel Frade, Elena Lopez-Guadamillas, Maribel Muñoz, Susana Llanos, Andrés Fernández, Aranzazu Sierra, Andrés Lopez, Noemí Haro and Ildefonso Rodríguez for their excellent technical and scientific support. Work at the laboratory of P.J.F.M. is funded by the Ramón Areces Foundation, (CIVP18A3891), Asociación Española contra el Cáncer-AECC (SIRTBIO—LABAE18008FERN), a Ramon y Cajal Award from the Spanish Ministry of Science, Innovation and Universities (MICINN) (RYC-2017-22335), RETOS projects Program of MICINN (SAF2017-85766-R) and the Portuguese Foundation for Science and Technology (FCT-MCTES, SFRH/BD/124022/2016). Work at the laboratory of ARM was funded by the MICINN (PID2019-110183RB-C21), Regional Government of Community of Madrid (P2018/BAA-4343-ALIBIRD2020-CM) and the Ramón Areces Foundation. Work at the laboratory of A.D.R. Funded by the Comunidad de Madrid—Talento Grant 2018-T1/BMD-11966 and the MICINN PID-2019-106893RA-100. Work at the laboratory of L.D. is fun-ded by projects from the Health Research Fund (ISCIII FIS PI14/01374 and FISPI17/00508) and from a Manuel de Oya research fellowship from the Beer and Health Foundation. Work at the laboratory of A.E. is funded by a Ramon y Cajal Award from MICINN (RYC-2013-13546) and RETOS projects Program of the MICINN, co-funded by the European Regional Development Fund (ERDF) (SAF2015-67538-R). Work in the laboratory of M.S. was funded by the IRB and by grants from the Spanish Ministry of Economy co-funded by the European Regional Development Fund (ERDF) (SAF2013-48256-R), the European Research Council (ERC-2014-AdG/669622), and the "laCaixa" Foundation.

## Author contributions

M.B. and A.P. performed most of the experiments, analyzed the data, and helped writing the paper. G.C. and J.H. oriented and helped with the biostatistical analysis. L.F.C.M., J.L.L.A., and C.P. helped with some of the experiments. V.M. and L.D. performed experiments and helped analyz-ing data for circulating miRNAs. R.M., H.T., and F.A.S. performed the bioinformatic analysis. V.L.K. coordinated the human nutritional assay and helped in data analysis. A.D.R. helped in the data analysis and interpretation. A.S.V. and I.L. analyzed erythrocyte membrane lipids and helped in the data analysis. O.J.P. performed the lipidomic profile ana-lysis. A.R.d.M. and M.S. collaborated in the human nutritional assay. A.E. collaborated in the RNAseq and the mouse experiments. P.J.F.M. designed the study, helped in some experiments and wrote the paper.

## Competing interests

The authors declare no competing interests.

## Additional information

[1]Metabolic Syndrome Group—BIOPROMET, CEI UAM+CSIC, Madrid Institute for Advanced Studies—IMDEA Food, Madrid, Spain. [2]Biostatistics and Bioinformatics Unit, CEI UAM+CSIC, Madrid Institute for Advanced Studies—IMDEA Food, Madrid, Spain. [3]Cardiovascular risk and nutrition, Hospital del Mar Medical Research Institute—IMIM, Barcelona, Spain. [4]Nutritional Genomics of Cardiovascular Disease and Obesity, Madrid Institute for Advanced Studies—IMDEA Food, Madrid, Spain. [5]Bioinformatics Unit, Spanish National Cancer Research Centre—CNIO, Madrid, Spain. [6]Nutritional Interventions Group, Precision Nutrition and Aging, CEI UAM+CSIC, Madrid Institute for Advanced Studies—IMDEA Food, Madrid, Spain. [7]Nutrition and Clinical Trials Unit, Platform GENYAL, CEI UAM+CSIC, Madrid Institute for Advanced Studies—IMDEA Food, Madrid, Spain. [8]Molecular Oncology and Nutritional Genomics of Cancer Group, CEI UAM+CSIC, Madrid Institute for Advanced Studies—IMDEA Food, Madrid, Spain. [9]Metabolism and Cell Signaling Group, Spanish National Cancer Research Centre—CNIO, Madrid, Spain. [10]Institute for Research in Biomedicine (IRB Barcelona), Barcelona Institute of Science and Technology (BIST), Catalan Institution for Research and Advanced Studies (ICREA), Barcelona, Spain. [11]Applied Metabolomics Research Group, Hospital del Mar Medical Research Institute—(IMIM), Barcelona, Spain. [12]Fatty Acid Research Institute, Sioux Falls, SD, USA. [13]These authors contributed equally: Marta Barradas, Adrián Plaza. ✉e-mail: marta.barradas@imdea.org; adrian.plaza@imdea.org; pablojose.fernandez@imdea.org

