## [Peer Review File · Nature Communications]

REVIEWER COMMENTS

Reviewer #1 (Remarks to the Author):

Barradas and colleagues collected blood samples from healthy human volunteers and mice before and after 36 or 24 hours of fasting, respectively, to measure fatty acid composition of erythrocyte membranes, circulating miRNAs, and RNA expression at PBMCs. In both mice and humans, fasting affected the proportion of polyunsaturated versus saturated and monounsaturated fatty acids at the erythrocyte membrane. Also, fasting significantly reduced the expression of insulin signaling-related genes. They show that 24 hours of fasting before and 24 hours after administration of the chemotherapeutic drug oxaliplatin in mice causes a strong protection from kidney, liver, heart and bone marrow toxicity. They provide data supporting two clearly separated groups of individuals that accurately predict a differential protection from chemotherapy toxicity thus potentially providing biomarkers to predict fasting-mediated protection from chemotherapy toxicity.

This is an excellent study providing important clues on how fasting can protect mice and possibly humans from chemotherapy toxicity.

The finding that the levels of specific PUFAs vs. SFAs and MUFAs change inversely with fasting is very interesting. Because reduced glucose and IGF-1 are well established markers of fasting and are at the center of the protective effects of fasting it would be important to include in the paper an analysis of the correlation of the changes in specific fatty acids and both glucose and IGF-1.

The fact that previous reports studied serum circulating miRNAs in humans after a short, overnight fasting, and found no differences and that the longer, 36-hour fasting, causes many circulating miRNAs changes in the cell-free human plasma is very interesting and should be emphasized in the discussion considering the importance of moving away from words like "fasting" and toward defining the exact period and type of fasting applied.

Line 368: Are the author proposing that insulin resistance and insulin levels are inversely correlated with the effects of fasting on biomarkers? This should be made clearer

The effects of fasting of chemotherapy damage shown in Fig.5 are remarkable and the authors have done an excellent job characterizing gene expression changes and toxicity in many organs/systems. Because the effects of fasting are different in different organs the authors should clarify in which organs/systems oxaliplatin is known to be toxic and in which is not or only causes a minor toxicity based on the literature and their own studies.

In figure 6 and earlier, the authors should clarify whether it is an increase or decrease in the insulin response genes that is associated with a lower toxicity in the fasted mice.

Minor

It would be helpful to more easily assess each experiment if the number of mice or human participants was included in each panel

Reviewer #2 (Remarks to the Author):

Barradas et al. describe the differences in fatty acid saturation on erythrocytes after fasting and the potential protection of normal cells from oxaliplatin toxicity with fasting. They studied 20 healthy volunteers and 26 mice with similar fasting protocols. Overall, this is an interesting descriptive paper about the effects of fasting in mice and humans. The utility of this data in the setting of chemotherapy is certainly limited given oxaliplatin is only one type of chemotherapy and is usually not given as a single agent for oncology.

Major comments/questions

1) Section on "new transcriptional regulation" – the authors write that there is decreased transcription, but this is probably more accurately represented as expression since you cannot determine if the changes are decreased transcription or increased RNA destruction/processing by miRNA or other factors.

2) I am having difficulty following why the authors chose to analyze fatty acid genes from the liver

in PBMCs. It would be surprising to see conserved changes of these genes between this different cell types. What about all other genes? Was there any genes that were concordant in mouse and human? I understand the authors are probably trying to focus analysis on lipid metabolism, but it does seem to be forced

3) Oxaliplatin is not usually given as a single agent and the toxicity of oxaliplatin that is most pronounced is neurotoxicity, which was not evaluated in the mouse. Would consider evaluating this as this is the most relevant for patients and also consider evaluating other chemotherapy to determine if fasting is more generalizable to multiple chemotherapies

Reviewer #3 (Remarks to the Author):

The authors use two separate studies on 20 human volunteers (25% obese) and a set mice, with different fasting regimes. Unfortunately, the fasting period for mice was 24h. Such long periods of time cannot be called short fasting, but indeed is already a long fasting period. Mice exhibit strong production of ketone bodies even after an 8h fast, very different to humans who have a maximum ketone production after 72h fasting. Indeed, this fact was also shown by the figures provided by the authors that even showed a drastic loss in body mass in mice after 24h (along with equally drastic responses in glucose, insulin, ketone bodies and other parameters). The reviewer therefore does not think that the two studies readily compared because in humans, such fast is still considered a short fast, but not in mice. The authors then studied changes in erythrocyte total fatty acid compositions which presumably may reflect changes in membranes in other organs. Such changes in other organs were not shown (for mice). Moreover, only proportional changes in fatty acid compositions were shown, but not absolute changes. It is therefore impossible to derive a meaningful "median" to possibly distinguish groups of humans who might show more, or less, toxicity to chemotherapy. The link between changes in fatty acid compositions to differences in chemotherapy responses were not convincingly demonstrated. While in humans, changes in miRNA were observed, these changes were not shown in mice. Changes in specific genes involved in lipid biosynthesis were not meaningfully related to chemotherapy response, either: of course, fasting leads to decrease in lipid biosynthesis. This is old news. However, the differences in PUFAs versus MUFA/SFA could be interesting, but was not explored mechanistically. Similarly, the authors chose to investigate total fatty acid profiles, instead of detailing changes in specific lipid membranes (such as PC) over other lipid membranes (such as plasmalogens). Similarly, it would have been interesting to study changes in fatty acyls in ceramides and sphingomyelins. Such details might have yielded functional links between chemotoxicity response and membrane stability or membrane fluidity. Currently, the authors did not convincingly demonstrate a functional relationship, but only statistical associations.

Response to Reviewer's comments.

[AUTHORS] We thank the reviewers for their time and careful evaluation of our manuscript. We believe that their comments have helped us to improve the quality of our paper. We have addressed all their comments, as explained below, including the addition of significant new data and modifications to the text. We apologize for the delay in our response, but different non-scientific problems have strongly hampered our work. We trust that the reviewers will now find our new manuscript appropriate for publication in *Nature Communications*.

We also thank the reviewers for their encouraging comments:

Reviewer #1: "This is an excellent study providing important clues on how fasting can protect mice and possibly humans from chemotherapy toxicity"; "The finding that the levels of specific PUFAs vs. SFAs and MUFAs change inversely with fasting is very interesting".

Reviewer #2: "Overall, this is an interesting descriptive paper about the effects of fasting in mice and humans".

[Reviewer #1]

Barradas and colleagues collected blood samples from healthy human volunteers and mice before and after 36 or 24 hours of fasting, respectively, to measure fatty acid composition of erythrocyte membranes, circulating miRNAs, and RNA expression at PBMCs. In both mice and humans, fasting affected the proportion of polyunsaturated versus saturated and monounsaturated fatty acids at the erythrocyte membrane. Also, fasting significantly reduced the expression of insulin signaling-related genes. They show that 24 hours of fasting before and 24 hours after administration of the chemotherapeutic drug oxaliplatin in mice causes a strong protection from kidney, liver, heart and bone marrow toxicity. They provide data supporting two clearly separated groups of individuals that accurately predict a differential protection from chemotherapy toxicity thus potentially providing biomarkers to predict fasting-mediated protection from chemotherapy toxicity.

This is an excellent study providing important clues on how fasting can protect mice and possibly humans from chemotherapy toxicity. The finding that the levels of specific PUFAs vs. SFAs and MUFAs change inversely with fasting is very interesting. Because reduced glucose and IGF-1 are well established markers of fasting and are at the center of the protective effects of fasting *it would be important to include in the paper an analysis of the correlation of the changes in specific fatty acids and both glucose and IGF-1*.

[AUTHORS]

We thank this reviewer for this interesting suggestion. We had already measured the plasma glucose and insulin levels in fasting volunteers. Actually, basal insulin, fasting insulin and Δ Insulin were among the plasma biomarkers most closely correlating with our new fasting biomarkers (membrane fatty acids, miRNAs and PBMC genes) in humans and mice, as shown in Table 3. Following this reviewer's suggestion, we have also measured the plasma levels of IGF-1 in our human samples, and have included these data in Table 3. Indeed, Δ IGF-1 is among the biochemical biomarkers that best correlate with our new fasting biomarkers, as shown in new Table 3. Supporting this Table, we have also included a new correlation plot in new Figure S4e and S4f, complementing the previous Figures 4c and 4d with the correlations of the best morphometric and biochemical parameters shown in Table 3 with erythrocyte membrane fatty acids.

Reviewer#1

The fact that previous reports studied serum circulating miRNAs in humans after a short, overnight fasting, and found no differences and that the longer, 36-hour fasting, causes many circulating miRNAs changes in the cell-free human plasma is very interesting and should be emphasized in the discussion considering the importance of moving away from words like "fasting" and toward defining the exact period and type of fasting applied.

[AUTHORS]

We are very happy that the reviewer finds this result interesting. Following the reviewer's suggestion, we have included a commentary about this finding in the Discussion section: "*This discrepancy between our study and that by Max and colleagues suggests that time of fasting is a critical parameter when studying physiological and molecular changes, and reinforces the importance of moving away from general words like "fasting" and toward defining the exact period and type of fasting applied*". (Lines 681-685).

Reviewer#1:

Line 368: Are the author proposing that insulin resistance and insulin levels are inversely correlated with the effects of fasting on biomarkers? This should be made clearer.

[AUTHORS]

We appreciate this comment by Reviewer#1. Indeed, insulin resistance (HOMA-IR) and insulin levels before and/or after a fasting period are directly associated with Δ SFAs and Δ MUFAs, and inversely associated with Δ PUFAs both in humans and mice, as shown in Table 3. We have included the new Figures S4e and S4f to make these correlations clearer, and have included a comment in their description, lines 429-432: "these results indicate that the insulin sensitivity measured before and after a fasting period is a strong predictor of our new fasting-triggered mechanisms, and strongly correlate in a coordinated manner with the evolution with fasting of the two groups of membrane fatty acids: Δ PUFAs vs Δ MUFAs/ Δ SFAs".

Reviewer #1:

The effects of fasting of chemotherapy damage shown in Fig.5 are remarkable and the authors have done an excellent job characterizing gene expression changes and toxicity in many organs/systems. Because the effects of fasting are different in different organs the authors should clarify in which organs/systems oxaliplatin is known to be toxic and in which is not or only causes a minor toxicity based on the literature and their own studies.

[AUTHORS]:

We have included this paragraph in the main text, lines 458-465:

"The most common oxaliplatin side effects, according to the existing literature⁵⁷⁻⁶¹, are: neurotoxicity, the main responsible for discontinuation of oxaliplatin treatment, even more than tumor progression⁵⁷; hematological toxicity, most commonly neutropenia (more than 1/3 of patients) and, more rarely, thrombocytopenia⁶⁰; liver damage, including sinusoidal obstruction and steatohepatitis⁵⁸; acute kidney damage⁵⁹; increased cardiotoxicity⁶¹; and gastrointestinal toxicity in the form of easily treated conditions such as diarrhea, vomits, nausea or mucositis⁶²".

Following this reviewer's comment, we have included new measurements of neurotoxicity: the von Frey assay and the expression of the genes *My11*, *Myh4*, *Tnnc2*, *Tnnt3* and *Cd74* in the dorsal root ganglia. As shown in Figure 5d, mice subjected to fasting were significantly protected from the increase in these neurotoxicity genes and from the increased paw sensitivity caused by oxaliplatin treatment. In addition, we have observed a marked thrombocytopenia, one of the hematological conditions elicited with oxaliplatin treatment, that was indeed prevented by fasting, as shown in Figure 5e. Neutropenia was also induced after oxaliplatin administration, although in this case fasting did not prevent this condition, as we show in the new Figure S5b; we have observed liver sinusoidal dilatation and accumulation of lipid vacuoles with oxaliplatin, summarized in the liver damage score shown in Figure 5f and h, along with several transcriptional biomarkers of liver damage; and we have measured kidney and heart toxicity using several transcriptional and histological biomarkers (Figure 5a, b, g, i).

In addition, and following the suggestions of several reviewers, we have also measured the effects of fasting in the toxicity elicited by doxorubicin, a chemotherapy unrelated to oxaliplatin used against lung, gastric, ovarian, breast, thyroid, sarcoma and pediatric cancers. The main doxorubicin secondary effect is cardiotoxicity⁸¹, followed by leukopenia⁸², hepatotoxicity⁸³ or renal toxicity^{84,85}. We have measured toxicity in these tissues after an intraperitoneal inoculation of 15 mg/kg of doxorubicin in 12-14-week-old male mice. Fasting protected from heart (Figure S5c), kidney (Figure S5e) and liver (Figure S5f) damages; and from leukopenia (most strongly to monocyte counts, but also showing a clear trend in total white blood cells and lymphocyte counts, Figure S5f), resulting in a much stronger recovery of body weight after chemotherapy administration, compared with their respective chemotherapy-treated fed controls (Figure S5g). We have indicated these findings in the main text, lines 483-494.

Reviewer #1

In figure 6 and earlier, the authors should clarify whether it is an increase or decrease in the insulin response genes that is associated with a lower toxicity in the fasted mice.

[AUTHORS]:

This is an important point that can give rise to confusion, and we thank the reviewer for pointing it out. Indeed, as shown in Figure 4a-b, insulin response genes *Srebf1*, *Scd1* and *Insig1* are downregulated with fasting and, therefore, $\Delta Srebf1$, $\Delta Scd1$, $\Delta Insig1$ are actually negative values. In Figure 6a, it is shown that higher values (that is: less negative) of the $\Delta Srebf1$, $\Delta Scd1$ and $\Delta Insig1$ insulin-responding genes, and of $\Delta Lpcat1$ and $\Delta Erg28$ (the insulin-related genes identified in our RNAseq experiment in Figure 4c-d), are associated with reduced chemotherapy toxicity. In other words: a smaller decrease in the expression of these genes with fasting is associated with lower chemotherapy toxicity, as is shown in Figures 6a and 7d for $\Delta Srebf1a$, $\Delta Scd1$ and $\Delta Insig1$. We have commented this issue in the Discussion section, lines 772-783.

Reviewer #1

It would be helpful to more easily assess each experiment if the number of mice or human participants was included in each panel.

[AUTHORS]:

Following the suggestion of this reviewer, we have indicated the number of individuals in each panel, when space was available; and, in the few cases when room was too scarce, we have indicated this information in the corresponding figure legend. We hope this relevant issue is clearer now.

Reviewer #2

The utility of this data in the setting of chemotherapy is certainly limited given oxaliplatin is only one type of chemotherapy and is usually not given as a single agent for oncology.

[AUTHORS]:

We acknowledge the importance of assessing the ability of fasting to protect from toxicity of chemotherapies different from oxaliplatin, following the suggestion of this reviewer. For this, we have treated mice with doxorubicin, another type of chemotherapy unrelated to oxaliplatin and with different toxicity profiles. We have shown that fasting protects from doxorubicin toxicity in the main affected tissues, including heart, lymphocytes, liver and kidney, as shown in new Figure S5c-g and explained in the text lines 483-494.

In addition, we want to stress that neurotoxicity, one of the most prevalent toxicities of the combined chemotherapies comprising oxaliplatin (oxaliplatin + fluorouracil or oxaliplatin + capecitabine), is specifically associated to oxaliplatin, and not to the other combined drugs; and fasting protects from this type of toxicity (Figure 5d, S5m and S6d), adding to the potential clinical translation of our results.

Reviewer #2

Section on “new transcriptional regulation” – the authors write that there is decreased transcription, but this is probably more accurately represented as expression since you cannot determine if the changes are decreased transcription or increased RNA destruction/processing by miRNA or other factors.

[AUTHORS]:

We agree with this comment, and have substituted “transcription” for “mRNA expression” throughout the text, when applied to these new mRNA biomarkers.

Reviewer #2

I am having difficulty following why the authors chose to analyze fatty acid genes from the liver in PBMCs. It would be surprising to see conserved changes of these genes between this different cell types. What about all other genes? Was there any genes that were concordant in mouse and human? I understand the authors are probably trying to focus analysis on lipid metabolism, but it does seem to be forced

[AUTHORS]:

We agree with the reviewer that gene expression between two very different tissues, as liver and PBMCs, does not necessarily need to coincide. However, there are reports that already showed coincident changes of mRNA expression during fasting between liver and PBMCs. For example, in Bowens et al., 2007, it was shown that PPAR α target genes *PLIN2*, *CPT1A* and *SLC25A20*, already shown to be upregulated with fasting in liver, were also upregulated in PBMCs with fasting. Actually, following this reviewer's suggestion, we have carefully revised the literature and have found a report showing decreased expression of *Srebf1* mRNA in rat PBMCs after a 14-hour fasting period (Oliver et al., Integrative Physiology 2013), confirming our observations. We have mentioned this reference in the text⁴⁶.

We have also measured the mRNA expression in fed and fasted mouse livers and PBMCs of the genes identified in our RNAseq experiment, and have observed that changes with fasting of the expression of the *Srebf1a* pathway genes *Srebf1*, *Scd1* and *Insig1* coincided between liver and PBMCs. In the case of the insulin-related genes identified in our RNAseq experiment, only changes with fasting of the expression of *Erg28* coincided between liver and PBMCs; while *Lpcat*, *Vav3* and *Inpp5a* increased with fasting in liver but decreased in PBMCs (Figure S4c). We describe these new data in lines 408-412. As this reviewer indicates, we have not performed a general analysis comparing mRNA expression regulation with fasting in liver and PBMCs, but we believe that, with these new data included, our lipid metabolism-centered approach is robust and biologically meaningful.

Reviewer #2

Oxaliplatin is not usually given as a single agent and the toxicity of oxaliplatin that is most pronounced is neurotoxicity, which was not evaluated in the mouse. Would consider evaluating this as this is the most relevant for patients and also consider evaluating other chemotherapy to determine if fasting is more generalizable to multiple chemotherapies

[AUTHORS]:

Following this reviewer's suggestions, we have measured the effects of fasting on oxaliplatin-induced neurotoxicity in mice, and on the toxicity profile of doxorubicin, another common chemotherapy unrelated to oxaliplatin.

We have found that fasting protects mice from oxaliplatin-induced neurotoxicity using two sets of biomarkers:

- The behavioral Von Frey assay, consisting of measurements of the pain reaction to physical pressure on the paw palms by filaments of increasing rigidity before and 5 days after oxaliplatin treatment, with or without fasting, shown in Figure 5d, panel to the right and Figure S6d.

- mRNA expression in dorsal root ganglia of genes associated to neurotoxicity, Figure 5d and S6d.

As shown in Figure 5d and S5m and S6d, both sets of biomarkers show that fasting protects from oxaliplatin neurotoxicity.

In addition, and following this reviewer's suggestion, we have measured the effects of fasting when combined with doxorubicin, an anthracycline unrelated with oxaliplatin and with a different toxicity profile. As shown in Figure S5c-g, fasting protected from several toxicity markers when combined with doxorubicin, including heart, liver and kidney damage, leukopenia and total body weight loss.

Reviewer #3

Unfortunately, the fasting period for mice was 24h. Such long periods of time cannot be called short fasting, but indeed is already a long fasting period. Mice exhibit strong production of ketone bodies even after an 8h fast, very different to humans who have a maximum ketone production after 72h fasting. Indeed, this fact was also shown by the figures provided by the authors that even showed a drastic loss in body mass in mice after 24h (along with equally drastic responses in glucose, insulin, ketone bodies and other parameters). The reviewer therefore does not think that the two studies readily compared because in humans, such fast is still considered a short fast, but not in mice.

[AUTHORS]:

This is indeed a very relevant point, and we have tried to address the similarities and differences between mice and humans in terms of fasting biomarkers.

As indicated by this reviewer, human volunteers reach their maximum blood beta-hydroxybutyrate levels (~3-4mM) after 72 hours of fasting, presenting ~1mM after 24 hours of fasting, and ~2mM after 48 hours of fasting (Fazeli et al., 2015)³⁵. According to this paper, human volunteers lost ~1kg after 24 hours of fasting, and up to ~4kg after 72 hours of fasting; considering the average body weight of volunteers, this body weight loss represented 1.3% of the total body weight after 24 hours fasting, and up to 5.2% after 72 hours of fasting. To evaluate how comparable are the mouse and human fasting processes, we measured blood ketone body and glucose levels in mice, as well as their body weight loss. We observed that blood ketone bodies and glucose actually changed rather similarly between humans and mice: b-HB reached ~1mM after 24 hours of fasting, and reached ~2.5 mM after 48 hours of fasting; while glucose levels did not go below 50 mg/dl even after 48 hours of fasting (Figure S1a-b). Body weight loss was more pronounced in mice than in humans, reaching >13% of BW loss at 24 hours, although mice recover this BW extremely fast after refeeding, as shown in Figure 5j and S5g. We have included these data in the new Figure S1a-c, and commented these results in lines 166-182. Based on this time course experiment and the available literature, we considered that 48 hours of fasting are comparable enough between mice and humans, and chose the 48 hours fasting protocol as our standard mouse fasting protocol for subsequent studies.

In addition, we have tested the ability of fasting periods shorter than 48 hours (24 hours) to protect from chemotherapy toxicity (Figure S5h-m). We observed that, although 24 hours of fasting tended to protect partially from oxaliplatin toxicity, it was not as effective as 48 hours of fasting. We have commented about these findings in lines 495-505 and 738-743.

Reviewer #3

The authors then studied changes in erythrocyte total fatty acid compositions which presumably may reflect changes in membranes in other organs. Such changes in other organs were not shown (for mice).

[AUTHORS]

Following this reviewer's suggestion, we have performed a fasting protocol and obtained erythrocytes, liver, kidney, heart and brain samples from fed control mice and 24-hours fasted mice. We have included these results in Figure S2c-h, and commented them in the main text, lines 222-254.

Reviewer #3

Moreover, only proportional changes in fatty acid compositions were shown, but not absolute changes. It is therefore impossible to derive a meaningful "median" to possibly distinguish groups of humans who might show more, or less, toxicity to chemotherapy.

[AUTHORS]

This is a good point. Fatty acid as proportions is still by far the most used way to express fatty acid data in body tissues (please see the landmark work by Hodson L and co-workers¹²⁰). However, we acknowledge that in the field of fatty acid research, there is a lack of consensus as to whether plasma fatty acids expressed as mol% (proportions) or concentration are more informative biomarkers, since both ways to report have significant caveats. Indeed, expressing fatty acid composition as a percentage of the total is meaningful only when determined fatty acid species are identical for all subjects. For this reason, in our experiments, we always measure the same fatty acids for all individuals. However, the use of absolute concentrations has substantial limitations, as well. These limitations mostly rely on the inter-individual differences of blood elements other than fatty acids (i.e.: lipoproteins, hydration, and hematocrit) that, if not taken into consideration, could lead to erroneous interpretations of the data. Hematocrit is key in our study, since (i) we are determining fatty acids in red blood cells; and (ii) fasting is a well-known factor affecting hematocrit (see Koscielniak et al., 2017). In addition, chemotherapy is also known to affect red blood cell count, as well²². Accordingly, assessment of the fatty acid absolute concentrations in red blood cells may generate data that are very difficult to interpret when individuals with different hematocrit are compared, and, in fact, any observed change of one individual fatty acid may be solely driven by the change of hematocrit, leading to data misinterpretation.

Reviewer #3

The link between changes in fatty acid compositions to differences in chemotherapy responses were not convincingly demonstrated.

[AUTHORS]

To prove that our results are reproducible, and to fulfill another request from this reviewer, we have repeated the same experiment shown in Figure 6, including a more complete lipidomic profile during fasting, and including new neurotoxicity measurements (new Figure S6). In this new experiment, we observed again a very tight association between changes with fasting of fatty acids and chemotherapy toxicity, in the same direction that we observed in our previous experiment in Figure 6a. We hope that this proof of reproducibility is convincing enough.

Reviewer #3

While in humans, changes in miRNA were observed, these changes were not shown in mice.

[AUTHORS] We acknowledge this caveat of our study. We hypothesize that mice may probably also alter other circulating different miRNAs in response to fasting, compared with what we observed in humans. Future research will reveal the exact identity of these altered miRNAs, their association to our new fasting biomarkers, and their relevance in the context of chemotherapy toxicity. Still, we consider that this caveat does not decrease the relevance and robustness of our findings in mice. We have commented on this issue in our Discussion section, lines 696-702.

Reviewer #3

Changes in specific genes involved in lipid biosynthesis were not meaningfully related to chemotherapy response, either: of course, fasting leads to decrease in lipid biosynthesis. This is old news.

[AUTHORS]

We have observed a very strong correlation between transcription of lipid biosynthesis genes *Srebf1*, *Insig1* and *Scd1* in PBMCs and the toxicity elicited by chemotherapy in different tissues, including kidney, liver, heart and bone marrow, as shown in Figure 6a. Moreover, when we measured these genes in the two groups of mice defined according to their response to fasting, there was a very significant difference between the two groups (Figure 7d and S7c), indicating that mice with a stronger decrease in the transcription of these genes were significantly less protected. Therefore, we consider that our results go way beyond a simple decrease in lipid biosynthesis genes, to a strong correlation between this change and many biomarkers of chemotherapy toxicity.

Reviewer #3

The differences in PUFAs versus MUFA/SFA could be interesting, but was not explored mechanistically. Similarly, the authors chose to investigate total fatty acid profiles, instead of detailing changes in specific lipid membranes (such as PC) over other lipid membranes (such as plasmalogens). Similarly, it would have been interesting to study changes in fatty acyls in ceramides and sphingomyelins. Such details might have yielded functional links between chemotoxicity response and membrane stability or membrane fluidity.

[AUTHORS]

We totally agree that a deeper lipidomic study would reveal more mechanistic insight into the functional relevance of the alterations in membrane composition with fasting, and its effects on chemotherapy toxicity. As a first attempt to answer this relevant question, we performed a lipidomic assay measuring three lipid families (ceramides, hexosyl-ceramides and lysophosphatidylcholines) in erythrocytes from basal and fasted conditions. These mice were then treated with chemotherapy, following the scheme shown in Figure 1b, and toxicity was measured by the biomarkers shown in Figure 5. We have included the correlation of all these new measurements in new Figure S6d, and present these results in lines 561-589.

Reviewer #3

Currently, the authors did not convincingly demonstrate a functional relationship, but only statistical associations.

[AUTHORS]

We agree that our findings are based on correlations between different parameters during fasting and chemotherapy. We also consider that these correlations have strong functional implications, since alterations in membrane composition and in the insulin signaling pathway are indeed exerting functional changes, as those shown for membrane-bound PLA2 and SMase activities in liver (Figure S2i-j). Even more importantly: since we can associate these correlations to chemotherapy toxicity, our findings have a strong functional application as biomarkers of fasting-mediated benefits, laying the foundation for future trials in the topic. We have included these comments in the Discussion section, lines 792-800.

REVIEWERS' COMMENTS

Reviewer #1 (Remarks to the Author):

the authors have addressed my concerns

Reviewer #2 (Remarks to the Author):

The authors have addressed my main critiques. Overall, this is an interesting publication demonstrating the importance of dietary changes in drug toxicity.

Reviewer #3 (Remarks to the Author):

The authors have satisfactorily answered the critiques. They have added new data to solidify their conclusions and carefully discussed the limitations, and prospects, of the study.

REVIEWERS' COMMENTS

Reviewer #1 (Remarks to the Author):

the authors have addressed my concerns

Reviewer #2 (Remarks to the Author):

The authors have addressed my main critiques. Overall, this is an interesting publication demonstrating the importance of dietary changes in drug toxicity.

Reviewer #3 (Remarks to the Author):

The authors have satisfactorily answered the critiques. They have added new data to solidify their conclusions and carefully discussed the limitations, and prospects, of the study.

We heartfully thank the reviewers for their positive insight and suggestions, that have significantly strengthened our results.